# Light-responsive and ultrapermeable two-dimensional metal-organic framework membrane for efficient ionic energy harvesting

Jin Wang [1] ✉, Zeyuan Song[1], Miaolu He[1], Yongchao Qian[2], Di Wang[1], Zheng Cui[1], Yuan Feng[1], Shangzhen Li[1], Bo Huang [3], Xiangyu Kong [2] ✉, Jinming Han[1] & Lei Wang [1] ✉

Nanofluidic membranes offer exceptional promise for osmotic energy conversion, but the challenge of balancing ionic selectivity and permeability persists. Here, we present a bionic nanofluidic system based on two-dimensional (2D) copper tetra-(4-carboxyphenyl) porphyrin framework (Cu-TCPP). The inherent nanoporous structure and horizontal interlayer channels endow the Cu-TCPP membrane with ultrahigh ion permeability and allow for a power density of 16.64 W m$^{-2}$, surpassing state of-the-art nanochannel membranes. Moreover, leveraging the photo-thermal property of Cu-TCPP, light-controlled ion active transport is realized even under natural sunlight. By combining solar energy with salinity gradient, the driving force for ion transport is reinforced, leading to further improvements in energy conversion performance. Notably, light could even eliminate the need for salinity gradient, achieving a power density of 0.82 W m$^{-2}$ in a symmetric solution system. Our work introduces a new perspective on developing advanced membranes for solar/ionic energy conversion and extends the concept of salinity energy to a notion of ionic energy.

Given the escalating global energy demand and concerns regarding the environmental impact of traditional energy sources, there is an urgent requirement to explore environmentally friendly and sustainable energy alternatives[1]. Osmotic energy, which is derived from electrolyte solutions with different salinities, has attracted significant attention as a promising solution due to its exceptional characteristics, such as abundant availability, cleanliness, and sustainability[2–4]. Reverse electrodialysis (RED) technology based on the selectively separates ions with different polarity with nanochannel membrane, is the most

promising approach for directly converting the Gibbs free energy of salinity gradients into electrical energy[5,6].

The exceptional ion perm-selectivity observed in biological protein channels within cell membranes offer significant inspiration for efficient ionic energy conversion[7,8]. A notable example is the electric eel, which harnesses the abundant nanoscale ion channels to convert ionic energy into electric shocks reaching approximately 600 volts[2]. To address the dual requirements of high permeability to counterions and strong selectivity for counter-ions similar to biological

[1]Research Institute of Membrane Separation Technology of Shaanxi Province, Key Laboratory of Membrane Separation of Shaanxi Province, School of Environmental & Municipal Engineering, Xi'an University of Architecture and Technology, No. 13 Yan Ta Road, Xi'an 710000, China. [2]CAS Key Laboratory of Bio-inspired Materials and Interfacial Science, Technical Institute of Physics and Chemistry, Chinese Academy of Sciences, No. 29 Zhongguancun East Road, Beijing 100190, China. [3]Institute of Chemical Engineering and Technology, Xi'an Jiaotong University, No. 28, West Xianning Road, Xi'an 710049, China. ✉e-mail: wangjin@xauat.edu.cn; kongxiangyu@mail.ipc.ac.cn; wl0178@126.com

**Fig. 1 | Schematic illustration of the preparation of two-dimensional (2D) copper tetra-(4-carboxyphenyl) porphyrin framework (Cu-TCPP) lamellar membrane with excellent photothermal effect.** The Cu-TCPP nanosheets were obtained by a modified thermal solvent method combined with sonication. Assembled by stacking nanosheets layer by layer to form a 2D Cu-TCPP lamellar membrane. The Cu-TCPP membranes with rich pores, multiple nanochannels, and photothermal conversion effect were obtained.

nanochannels, numerous strategies have been proposed for the structure design of bionic nanochannel membranes[9,10]. In contrast to one-dimensional (1D) nanopores/channels with complex production process, the two-dimensional (2D) lamellar membrane, easily prepared by parallel stacking of ultra-thin nanosheets, exhibits promising potential for osmotic energy harvesting[8,11,12]. Until now, many efforts have been made to optimize the properties of nanochannels, including their geometric characteristics and charge properties, to enhance ion selectivity and permeability[13,14]. Additionally, the significant impact of RED device design on ion transport behavior has also been reported. For example, the vertical model, where ions travel perpendicularly through the 2D nanochannels, has been the predominant choice for evaluating energy conversion efficiency. However, recent advancements in research have shown that when ions are transported parallel to the 2D nanochannels, a higher ion diffusion rate and improved RED performance are achieved due to shortened ion transport paths and minimized kinetic energy dissipation[15,16]. Recent studies have also introduced 2D MOF nanosheets with intrinsic, regular, and uniformly sized pores for constructing MOF-layered membranes[17]. Leveraging both the in-plane pores and interlayer transport pathways, these 2D MOF membranes have demonstrated considerable promise in gas and molecular separation[18–21]. However, the utilization of 2D MOF membrane in osmotic energy conversion applications remains notably scarce.

In addition to nanostructure of biological nanochannels, researchers have also drawn significant motivation from the stimuli-response ion transport properties of biological systems to enhance osmotic energy harvesting performance[7,22]. By now, external environmental factors such as pH, voltage, light have been demonstrated to regulate the ion diffusion process. For instance, the ion selectivity of Cu-TCPP nanochannels can be controlled by adjusting the surface charge density through pH modulation in the external environment[23]. In graphene-based lamellar membranes, ion diffusion can be controlled by directly applying an external voltage[24]. Furthermore, light with multiple advantages such as abundance, safety, low cost, and widespread availability has emerged as a significant method for modulating ion transport[25]. Utilizing the photothermal properties of MXene, Luo et al. demonstrated the rapid switching of ionic conductivity by controlling the evaporation of water within the MXene nanofluidic channel under irradiation[26]. Guo et al. observed active ion transport diffusion across WS$_2$-based 2D nanochannel membranes attributed to the separation of electron-hole separation under light irradiation[27].

In this study, we propose a nanofluidic system based on a 2D copper tetra-(4-carboxyphenyl) porphyrin (Cu-TCPP) nanosheet to achieve high ion sieving ability and stimulus-response transport properties (Fig. 1). The Cu-TCPP nanofluidic membrane with abundant horizontal pathways and vertical shortcuts for selective ion transport demonstrated an impressive power density of 16.64 W m$^{-2}$ under artificial seawater/river water conditions, surpassing that of state-of-the-art nanochannel membranes. Further enhancement of the energy output could be achieved by connecting the single devices to power a calculator and LED lamp. Furthermore, the exceptional photothermal conversion efficiency of the Cu-TCPP nanosheet enables light-controlled active ion transport. Consequently, the maximum power density in the seawater/river system increased to 31.92 W m$^{-2}$, nearly doubling the initial value. Moreover, even in the absence of salinity gradients, a power density of 0.82 W m$^{-2}$ was achieved, such performance greatly expanded the potential applications of ionic energy conversion.

## Results
### Characterization of Cu-TCPP nanosheets and lamellar membranes
Bulk Cu-TCPP crystals were synthesized via the optimizing solvent-thermal method where pyrazine used as a cofactor[28]. Scanning electron microscopy (SEM) revealed a loose and layered crystalline structure of bulk Cu-TCPP due to a higher growth rate in the horizontal direction than in the vertical direction during crystallization (Supplementary Fig. 1). The UV-Vis absorption spectroscopy tests of bulk

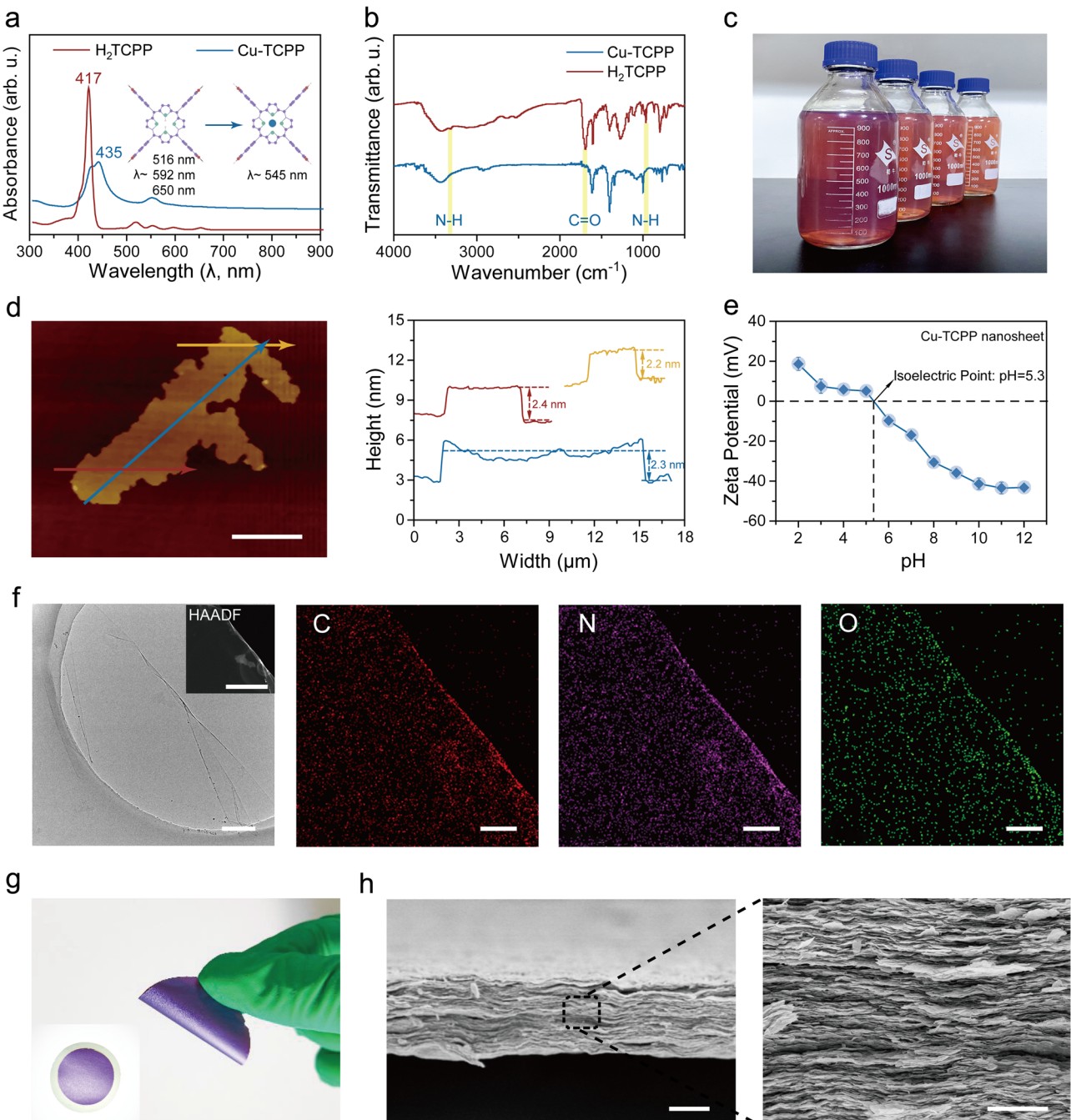

**Fig. 2 | Characterization of nanosheets and Cu-TCPP lamellar membranes. a** UV-Vis spectra of H₂TCPP and Cu-TCPP. The schematic diagram showed the replacement of hydrogen atoms by copper atoms in H₂TCPP to form Cu-TCPP. **b** FT-IR spectra of Cu-TCPP and H₂TCPP. **c** Image of prepared nanosheet in ethanol dispersions. **d** AFM image of individual Cu-TCPP nanosheet. The scanning direction is the direction of the arrow in the image. Scale bar, 3 μm. **e** Zeta potential of Cu-TCPP nanosheet dispersions versus pH value. **f** TEM images of Cu-TCPP nanosheets and the corresponding elemental mapping images of C, N, and O. Scale bar is 200 nm for HAADF image and 1 μm for other images. **g** Optical image and flexibility demonstration of Cu-TCPP membrane. **h** SEM image of the cross-sectional morphology of the Cu-TCPP membrane. Scale bar, 10 μm (Left). Scale bar, 2 μm (Right).

H₂TCPP solution and Cu-TCPP dispersions showed that the peaks at 516, 592, and 650 nm, originally presented in the spectrum of H₂TCPP, disappeared in that of Cu-TCPP, indicating the complete ligand-metal ion reaction (Fig. 2a). Moreover, the maximum absorption peak at 435 nm and the clear Q-Band absorption peak at 545 nm in the spectrum of Cu-TCPP confirmed that copper ion occupied the center of the H₂TCPP molecule[23]. Pyrazine, which assists the metal ions in binding to the ligand, was almost absent in the Cu-TCPP nanosheet dispersion after thorough washing (Supplementary Fig. 2)[29]. Fourier transform

infrared (FT-IR) spectra also showed that the N-H stretching vibration at 3315 cm⁻¹ disappeared, the in-plane N-H vibration at 964 cm⁻¹ decreased, and a new Cu-N absorption peak appeared at 1000 cm⁻¹, indicating the substitution of the hydrogen proton by a copper ion (Fig. 2b). Two new peaks at 1620 and 1400 cm⁻¹ corresponded to C-O-Cu bonds, confirming the coordination of the carboxyl group of H₂TCPP with copper ions. The peak at 1700 cm⁻¹ almost disappeared, suggesting the decreasing C = O stretching vibration and the formation of Cu₂(COO)₄ paddlewheel metal nodes[30,31]. Based on the X-ray

photoelectron spectroscopy (XPS) tests results of Cu-TCPP membranes (Supplementary Fig. 3), the C 1s spectrum showed fitted peaks at about 284.8, 286.2, and 288.3 eV, representing the C-C, C-N, and C = O/C-O bonds. In addition, the high-resolution Cu 2p and O 1s spectra showed the presence of $Cu_2(COO)_4$ nodes. The binding energy peak of 944.3 eV in the Cu spectrum corresponded to the characteristic peak of $Cu^{2+}$; peaks centered at 934.8 and 954.7 eV indicated the coexistence of two Cu valence states ($Cu^+$ and $Cu^{2+}$). In addition, the N 1s XPS spectrum demonstrated at 398.6 eV and 400.3 eV, representing N-Cu and N-H peaks, confirming the coordination of the copper ion to the nitrogen atom in the center of the porphyrin ring[23,32].

High-efficient exfoliation of the obtained bulk crystals into ultrathin nanosheets via facile ultrasonic treatment was achieved because the coordination bonds between copper ions and carboxyl groups in the horizontal direction exhibited significantly higher strength compared to interlayer interactions such as hydrogen bonds and van der Waals forces(Fig. 2c)[28,30]. Previous studies have indicated that the theoretical thickness of a monolayer Cu-TCPP nanosheet is 0.27 nm[33]. However, the average thickness of the prepared Cu-TCPP nanosheet, as determined via atomic force microscopy (AFM), measured approximately 2.3 nm (Fig. 2d and Supplementary Fig. 4). The precision of AFM measurements could be influenced by adsorbed surfactants on both flakes and substrates[34]. However, even after accounting for errors caused by surface adsorption, the thickness of the Cu-TCPP nanosheets we prepared far exceeded their theoretical thickness. Hence, we conclude that the nanosheets prepared in this study are multi-layered rather than truly single-layered nanosheets. The Wide-Angle X-ray Scattering (WAXS) analysis was conducted to further elucidate the structural characteristics of the obtained multi-layer Cu-TCPP nanosheet (Supplementary Figs. 5 and 6). In the out-of-plane WAXS pattern of the MOF membrane, the peak at 19°, which also appeared in the XRD pattern of bulk Cu-TCPP crystal reported previously[23,35], corresponded to a d-spacing of 4.7 Å between the single-layer nanosheets within the multi-layered nanosheets. Moreover, the presence of two distinct peaks at $2\theta = 9°$ and 12°, absent in the Cu-TCPP crystal, indicated the d-spacing of the stacked multi-layer nanosheets as 0.97 nm and 0.73 nm, respectively. Similar findings have been reported in previous studies on 2D MOF membranes. For instance, in Yang et al.'s research, the XRD pattern of the 2D $Zn_2(bim)_4$ MOF membrane exhibited peaks representing the ordered pristine bulk $Zn_2(bim)_4$ crystal structure, along with peaks at low angles due to the expanded restacking of nanosheets[36]. Based on these observations, the size between multi-layer nanosheets, calculated by subtracting the theoretical thickness of one layer of Cu-TCPP, have an approximate diameter of 0.17 nm, presenting a significant challenge for ions to achieve transmembrane diffusion within this confined space. However, the nanochannels located between the multi-layer nanosheets, with channel size of approximately 0.6 nm, offer the necessary space for rapid ions transport. Additionally, the in-plane XRD results indicated that our few-layer Cu-TCPP was established through the misaligned stacking of monolayers. The adjacent monolayers were shifted by 1/4 of a unit cell along the a-axis, leading to the formation of pores with the effective size of 0.84 nm (Supplementary Fig. 6)[23,30,37]

Transmission electron microscopy (TEM) images also demonstrated that the Cu-TCPP nanosheets are quite thin, nearly transparent, and free of defects (Fig. 2f). Moreover, the energy-dispersive X-ray spectroscopy (EDS) results confirmed uniform dispersion of the C, N, O, and Cu atoms in the prepared few-layer Cu-TCPP nanosheet (Supplementary Fig. 7). In neutral conditions, the zeta potential of the Cu-TCPP nanosheets stabilized at −17 mV, indicating a negatively charged surface (Fig. 2e). As illustrated in Supplementary Fig. 8, nanosheets' inherent negative charge characteristics nearly remained consistent in various electrolyte solutions. In the presence of a $CaCl_2$ solution, the surface negative charges exhibited a modest decrease, which could be attributed to the pronounced screening effect exerted by the divalent

$Ca^{2+}$ ions. The Cu-TCPP nanosheet dispersion exhibited excellent dispersity and stability, with a distinct Tyndall effect (Supplementary Fig. 9). The dispersions remained stable with no clear settling or agglomeration observed over 30 days (Supplementary Figs. 10 and 11).

The Cu-TCPP nanosheets were assembled into a lamellar membrane on a PVDF porous substrate using a vacuum-assisted filtration approach (Supplementary Fig. 12). The free-standing Cu-TCPP membrane, which was peeled off from the substrate after thoroughly drying at room temperature, exhibited good flexibility (Fig. 2g). The Cu-TCPP membrane surface exhibited distinct wrinkles with no discernible structural defects (Supplementary Figs. 13–15). The large lateral size of Cu-TCPP nanosheets led to strong interlayer interaction between the parallel nanosheets[38], resulting in a homogeneous laminar structure, as shown in the cross-sectional SEM images (Fig. 2h and Supplementary Figs. 16 and 17). By controlling the loading of Cu-TCPP nanosheets, the thickness of the membranes could be precisely tuned from nanometers to micrometers (Supplementary Fig. 18). The water contact angle on the Cu-TCPP membrane surface was stabilized at 71°, indicating its hydrophilic surface (Supplementary Fig. 19).

## Osmotic power conversion performance of Cu-TCPP membrane

As discussed above, the intrinsic pore structure of Cu-TCPP nanosheets offers additional channels for selective ion transport. To further enhance the ion permeability and minimize kinetic energy dissipation, we implemented a horizontal transport model in our nanofluidic system. The lamellar Cu-TCPP membrane was cut into a rectangular shape and sealed by polydimethylsiloxane (PDMS) prepolymer, enabling the diffusion of ions solely through the Cu-TCPP membrane[26,39]. To increase the contact area between the lamellar membrane and the electrolyte solutions, a section of the Cu-TCPP membrane was also kept in the reservoir. The total membrane area was divided into three equal zones: Zones-I, II, and III, with corresponding reservoirs designated as Reservoir-I and Reservoir-III (Fig. 3a). A pair of Ag/AgCl electrodes were utilized to measure ion currents with scanning voltages, and the ionic conductance was determined from the slope of the current-voltage (I−V) curve. As illustrated in Fig. 3b, the conductance exhibited a linear relationship with concentration at high KCl concentrations, similar to that of bulk KCl solution. However, at lower concentrations, conductance deviated significantly from bulk behavior which was primarily attributed to the substantial influence of surface charges of the nanoconfined channel on the distribution of ions within it.

According to classical Electrical Double Layer (EDL) theory, an EDL region forms at the interface between the charged channel wall and the ionic solution due to electrostatic attraction to counterions and repulsion from co-ions. The Debye length ($\lambda_D$), characterizing the decay of electrostatic forces as a result of screening the surface charges with counterions in electrolyte solution, typically ranges from tens of nanometers down to Angstroms as per the Poisson-Boltzmann theory[40]. In micrometer-sized channels, the EDL thickness is negligible compared to channel dimensions, making the effect of surface charge potential on ion transport behavior less evident. However, based on our XRD and TEM results described above, the interlayer channel diameter in the Cu-TCPP membrane was confined to the nanometric scale. In this scenario, the spaces occupied by EDLs in the extremely confined Cu-TCPP channel could not be neglected, especially at low ionic concentrations. As a result, ion population and distribution in the nanochannel were strongly influenced by the channel surface charge, and the Cu-TCPP membrane exhibited outstanding separation ability towards cations and anions. Similar behavior had been observed in other 2D nanofluidic systems, suggesting the excellent cation/anion separation ability of Cu-TCPP membrane[2,9,39,41]. Additionally, we conducted immersion experiments to further illustrate the membrane's ion selectivity. The Cu-TCPP membrane was immersed in a 0.5 M KCl solution, after which its surface was subjected to EDS scanning

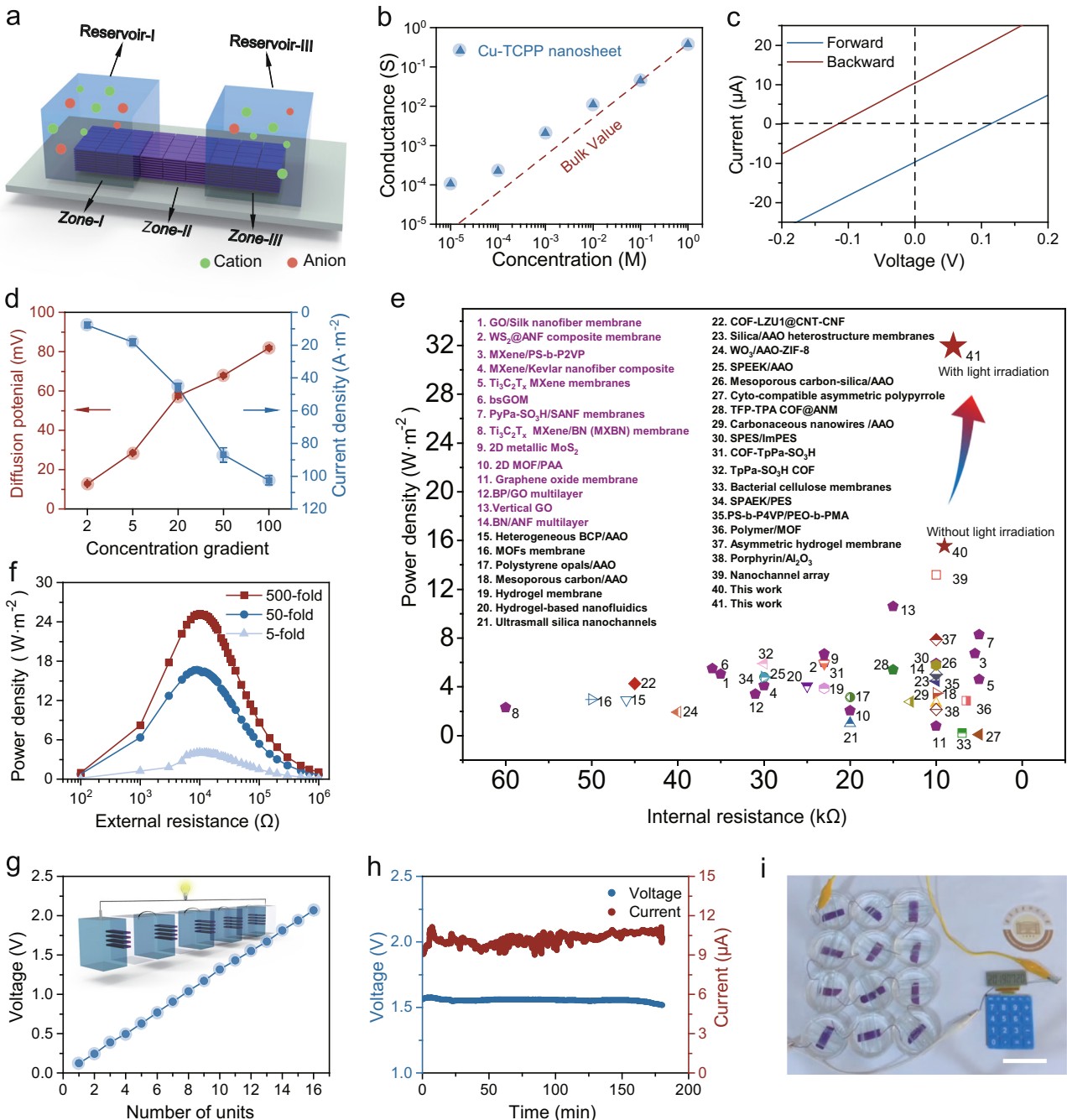

**Fig. 3 | High-performance osmotic energy conversion of Cu-TCPP membrane.**
**a** Scheme of the ionic energy conversion device. **b** Ionic conductivity as a function of KCl concentration. **c** I−V curves for the Cu-TCPP membrane tested in a 50-fold KCl concentration. **d** Open-circuit potential and short-circuit current of the Cu-TCPP membrane with the increasing concentration gradient. Error bars indicated the standard deviations from three different samples. **e** Comparison with previous studies under an artificial seawater/river water system (1st−14th purple color represents 2D laminar membranes, detailed and references for all data points are

found in Supplementary Table 2). **f** Output power density of Cu-TCPP membranes as a function of load resistance at different salinity gradients. **g** Linear relationship between voltage and number of the Cu-TCPP-RED device. (Insert Cu-TCPP-RED unit cells connected in series for the formation of a stacked device). **h** I-T and V-T curves of Cu-TCPP-RED devices for artificial seawater/river water system. **i** Twelve Cu-TCPP-RED unit cells connected in series produce a stable voltage for calculator to work. Scale bar, 5 cm.

(Supplementary Fig. 20). The higher adsorption amount of K⁺ compared to Cl⁻ on the Cu-TCPP membrane's surface further confirmed that the negatively charged Cu-TCPP nanochannels selectively attracted counterions due to electrostatic forces.

On this basis, the potential of Cu-TCPP nanochannel membrane in osmotic energy conversion applications was further investigated. To simulate the conditions of a seawater/river system, the NaCl solution concentrations in Reservoir-I and Reservoir-III were set at 0.5 M and

0.01 M, respectively. The I−V curves indicated clear short-circuit current ($I_{sc}$) and open-circuit voltage ($V_{oc}$) values of −125 mV and 9.73 μA, respectively (Fig. 3c). Here, under the driven of salinity gradient, both anions and cations tended to transport across the membrane from Reservoir-I to Reservoir-III. However, the cation selectivity of Cu-TCPP nanochannels restricted transmembrane diffusion to cations instead of anions, enabling the direct conversion of osmotic energy into electrical energy. Under a reverse concentration gradient, the $V_{oc}$ and $I_{sc}$ values

exhibited similar magnitudes but opposite directions, indicating the symmetry of the Cu-TCPP membrane structure (Fig. 3c).

Furthermore, to evaluate the relationship between salinity and energy conversion performance, the concentration of Reservoir-I was set to 1 M, while the concentration of Reservoir-III was varied from 0.01 to 0.5 M. To measure the diffusion potential ($E_{diff}$) and diffusion current ($I_{diff}$), we employed a pair of agar-saturated potassium chloride salt bridges to eliminate the electrode potential imbalance generated by the unequal potential drop at the electrode-solution interface (Supplementary Fig. 21). As illustrated in Fig. 3d, the $E_{diff}$ and $I_{diff}$ show a clear increase as the concentration gradient increased from 2 to 100 times. Furthermore, cation transfer number ($t_+$) and energy conversion efficiency ($\eta$) were further calculated as presented in Supplementary Fig. 22 and Table 1. The value of $t_+$ was consistently greater than 0.85 under different concentration gradients. Moreover, we observed that energy conversion efficiency ($\eta$) remained relatively constant and was not substantially altered as the concentration gradient increased, and the Cu-TCPP membrane was still capable of harvesting 29.2% of the energy in artificial seawater/river water systems.

The Cu-TCPP-based nanofluidic device was also integrated with an external load resistor ($R_L$) to form an external circuit. As depicted in Supplementary Fig. 23, the current decreased substantially with increasing load resistance in the artificial seawater/river water systems. The output power density reached a peak value of 16.64 W m$^{-2}$ when the loaded resistance was close to the internal resistance. To the best of our knowledge, this high power density at a 50-fold salinity gradient, not only surpassed the commercialization benchmark (~5 W m$^{-2}$) but also outperformed state-of-the-art nanochannel membranes as shown in Fig. 3e. Based on the above analysis of Cu-TCPP structure, we infer that this exceptional energy recovery capability resulted from both the horizontal ion transport model of the nanofluidic system constructed in our study and the unique structural features of the Cu-TCPP nanochannel itself[15,19]. Specifically, unlike vertically oriented 2D laminar structures, lateral transport in the Cu-TCPP nanochannel allowed for more entrances for ions to enter the channel[16]. The previous studies also demonstrated that ions undergo faster transport across the membrane with less consumption of kinetic energy during lateral transport, resulting in higher permeation rates both at the channel entrance and inside the channel[16,19]. Moreover, the nanochannel membrane built using 2D materials such as carbides and nitrides (MXene), transition metal dichalcogenides (TMDs), and vermiculite (VMT) suffer from a lack of internal transport channels due to the dense packing of atoms[42]. Despite containing a six-membered carbon ring, the sub-nanopores in the graphene have also been proven to be impermeable to ions[43]. In contrast, the Cu-TCPP nanosheet has a large number of regularly spaced nanopores, which provide more "short-cuts" for ion and water transport[18,19]. As a result, once ions entered the horizontal nanochannel, they underwent continuous transport instead of a tortuous path, allowing them to complete transmembrane diffusion within a shorter distance, thereby resulting in a high ionic current and excellent osmotic energy recovery performance as shown in Fig. 3e.

Experimental investigations were conducted to validate that the horizontal model is indeed more effective for ion transport than the vertical model. Under nearly identical salt gradients and a similar test area, even though the membrane length in the horizontal model (3 cm) was significantly longer than the membrane thickness in the vertical model (1 μm), the current in the horizontal model nearly matched that of the vertical model, indicating lower resistance in ion transmembrane transport in the horizontal model (Supplementary Figs. 24 and 25). Furthermore, we employed the Poisson-Nernst-Planck (PNP) equations to compute net ion currents across the cross-section under salinity gradient for both transport modes. A comprehensive set of parameters for both models were provided in Supplementary Figs. 26, 27, and Table 3. An increase in concentration on the reservoir-I side led to a higher net current produced by the horizontal model compared to the vertical model.

In previous studies, the nanochannel energy conversion efficiency was greatly hampered by diminished selectivity in a high-salinity environment. To explore if the Cu-TCPP membrane could maintain its excellent performance under high-salinity gradient conditions, we measured the power density after fixing the low-concentration NaCl solution at 0.01 M and varying the high-concentration NaCl from 0.05 to 5 M. Notably, as the concentration gradient increased from 5-fold to 50-fold to 500-fold, the power density value surged from 4 to 16.64 to 25.54 W m$^{-2}$, respectively, indicating the application potential of Cu-TCPP in the hypersaline environment (Fig. 3f and Supplementary Fig. 28).

Currently, in the field of osmotic energy harvesting, the inverse relationship between power density and increasing testing area is a critical challenge[44]. For the Cu-TCPP membrane, as the testing area increased from 0.01 to 0.2 mm$^2$, there was a sharp decrease in output density (Supplementary Fig. 29 and Table 4). As the testing area further increased, similar to the previous reports, the trend of the power density becomes more gradual. Previous reports[45,46] proposed that the relationship between the power density and the testing area could be demonstrated by the following equation:

$$\rho_m = \frac{P_m}{S} = \frac{1}{4} \times \frac{(1-\alpha)^2 U_0^2}{(R_I + R_{II})S + K_m} \tag{1}$$

In the equation, $\rho_m$ is the maximum power density, $S$ represents the test area, $U_O$ represents the open-circuit voltage, $\alpha$ is the loss coefficient due to the polarization potential, $R_T$ is the internal resistance of the test device, and $K_m$ represents the ion transport resistance factors across the gradient membrane. Based on these studies, under a fixed concentration gradient, $R_I$ and $R_{II}$ are kept constant for a given testing device. As a result, except for the $U_O$, and $K_m$ determined by ion-selective membranes, the output power of the RED device inversely correlates with the testing area, ultimately stabilizing at a low value. Moreover, Gao et al. have indicated that the concentration polarization phenomenon, caused by the substantial accumulation of K$^+$ ions at the channel exit as the testing area expands, will occur during the energy conversion process. Consequently, the reservoir/membrane interfacial resistance should not be ignored, leading to a decrease in power density with an increasing test area[47]. Elimelech et al. suggested that the unavoidable concentration polarization would also result in a decreased membrane potential and ionic conductance due to the lower transmembrane concentration gradient and decreasing driving force for ion diffusion[48]. It should be noted that the power density of Cu-TCPP membrane stabilized at 3.9 W m$^{-2}$ with the testing area of 0.6 mm$^2$, which is superior to most of the reported power density achieved with membranes of the same concentration difference and testing area. Furthermore, with the assistance of light irradiation, the ion diffusion driven by the synergistic effect of photothermal/concentration gradient led to a further increase in the power density of the Cu-TCPP membrane. When the test area reached 0.6 mm$^2$, the power density of the system achieved 8.27 W m$^{-2}$, surpassing the commercialization benchmark of 5 W m$^{-2}$ (Supplementary Fig. 30).

To assess the scalability of our Cu-TCPP membrane-based device and extend the osmotic energy recovery system, various numbers of Cu-TCPP membranes were assembled into an enlarged experimental setup. As depicted in Supplementary Fig. 31, a single Cu-TCPP-RED demonstrated a short-circuit current of 17.2 μA in an artificial seawater/river setup. For the Cu-TCPP-RED devices comprising 1 to 8 membrane units, the output current exhibited nearly linear growth in correlation with the number of membrane units, suggesting an increase in total ion flux with the expanded ion transport area. The output current also demonstrated excellent stability with negligible fluctuations over a duration of 7200 s (Supplementary Fig. 32). Furthermore, the power

density remained relatively stable despite the increase in the number of membrane units in the system, affirming the practical application potential of 2D Cu-TCPP membranes in osmotic energy harvesting.

A stack of series connected Cu-TCPP-based REDs was constructed to examine the scalability of energy harvesting performance from a single device to practical applications (Fig. 3g insert). The single Cu-TCPP-RED device, as described above, exhibited open-circuit voltages of 125 mV and short-circuit currents of 9.73 μA in artificial seawater/river system. A nearly perfect linear relationship was observed between the output voltage and the number of units in the Cu-TCPP RED stacks, which consisted of 1 to 16 units connected in series (Fig. 3g). The output current and voltage exhibited stability over an extended period with negligible variations (Fig. 3h). These results signified the performance consistency and operational stability of the Cu-TCPP-RED device (Supplementary Fig. 33), and it could be used to power a calculator (Fig. 3i and Supplementary Movie 1) and LED (Supplementary Fig. 34) for a long time. Real seawater from Bohai Sea (China) was also used to evaluate the osmotic energy conversion performance of the Cu-TCPP membrane under practical conditions (Supplementary Table 5). The power density of Cu-TCPP membrane reached 13.04 W m$^{-2}$ under actual seawater conditions (Supplementary Fig. 35).

Large-scale uniform Cu-TCPP membranes with an area of 78.5 cm$^2$ were successfully prepared by enlarging the filtration area (Supplementary Fig. 36), and the membrane structure remained intact even after being immersed in electrolyte solutions with different pH values for 15 days (Supplementary Fig. 37). Inductively coupled plasma (ICP) spectrometry was employed to monitor the Cu$^{2+}$ content in the solution during the immersion process of Cu-TCPP membranes. Remarkably, even after prolonged immersion for up to 15 days, the Cu$^{2+}$ content in the solution remained below the detection limit 0.5 mg L$^{-1}$. Furthermore, no disintegration or re-dispersion was observed even under 30 min of ultrasonic treatment (Supplementary Fig. 38), indicating that the Cu-TCPP membrane could withstand harsh working conditions. It is important to note that the production efficiency of the vacuum filtration method is quite low and time-consuming. To overcome this limitation, we employed the spraying coating strategy to fabricate a Cu-TCPP membrane with a larger area of 600 cm$^2$ at high speed (Detailed conditions described in the method and Supplementary Fig. 39), further highlighting the potential for large-scale production and industrial applications of the 2D Cu-TCPP membrane. A cost comparison between the Cu-TCPP membrane and other commercial membranes utilized in RED was also conducted (Supplementary Table 6). Considering the widely used Cu-TCPP membrane with a thickness of 4 μm in our experiments, the cost per square meter is approximately 300 RMB, which is at least 5 times lower than the price of commercial membranes.

## Photo-responsive ion transport behavior

To examine the light-driven ion transport behavior of Cu-TCPP nanochannels, we incorporated a Xenon lamp as the light source in the nanofluidic system (Fig. 4a). When the same concentration of NaCl solutions was filled into both reservoirs, no clear ionic current was detected under non-illuminated condition. However, upon irradiation of Zone-I, we observed a remarkable increase in ionic current, reaching −750 nA within 60 s. Upon cessation of light irradiation, the current promptly returned to its initial state, suggesting that the observed current and voltage were photo-induced (Fig. 4b, c). The photo-induced ionic current remained stable even during extended testing, and a corresponding photo-responsive voltage of 18 mV was also verified.

To investigate the underlying mechanism of light-induced ion transport, we utilized an infrared thermographic camera to monitor the temperature changes of the Cu-TCPP membrane before and after illumination (Fig. 4d and Supplementary Fig. 40). The findings

illustrated the excellent photothermal conversion performance of the Cu-TCPP membrane, as indicated by a swift temperature rise from room temperature to around 67 °C within 15 s of illumination. Upon prolonged irradiation for 90 s, the temperature continued to increase and eventually stabilized at around 120 °C. Based on these observations, we inferred that the photo-induced ionic current arises from the temperature gradient resulting from the photothermal properties of Cu-TCPP. Furthermore, the direction of the ionic current dependent on the temperature gradient created by the illumination of different membrane zones. In addition, illumination of Zone-I and Zone-III resulted in the formation of opposing temperature gradients, giving rise to currents and voltages of equal magnitude but opposite directions. Illumination of Zone-II generated opposing temperature fields on both sides of the membrane, causing the cancellation of the two currents, resulting in no significant photo-induced current or voltage. As a control experiment, no measurable temperature change was observed on the surface of the PVDF membrane after illumination (Supplementary Fig. 41), and no net ion photocurrent was detected.

In addition to the lower transport resistance in the vertical model, the horizontal model also provided the possibility to amplify the temperature gradient within the nanochannel, creating a more significant temperature difference to enhance the impact on ion transport efficiency. To validate this mechanism, we monitored temperature changes in Cu-TCPP membranes under both horizontal and vertical transport modes in the presence of light. In the horizontal model, one end of the membrane was illuminated, and the temperature difference by comparing the illuminated section with the opposite end was measured. In vertical model, we illuminated the front surface of the membrane and measured the temperature difference between the illuminated side and the opposite side of the membrane (Supplementary Fig. 42). The temperature difference in the Cu-TCPP membrane increased to 120 °C in just 90 s in the horizontal transportation model, whereas in the vertical transportation model, under the same irradiation conditions, the temperature difference was only 5 °C. Similarly, the PNP equation was utilized to compute the net ion currents in the cross-section for both transport modes under temperature gradient (Supplementary Fig. 27). When the concentration gradient of the nanofluidic system remained constant on both sides of the reservoir, an elevation in temperature on the reservoir-III side resulted in a considerably higher net current for the horizontal model than for the vertical model.

From a thermodynamic perspective, when partial photo-irradiation was applied to the Cu-TCPP membrane, the photothermal conversion effect decreased the Gibbs free energy in the illuminated region[49]. As a result, ions tended to migrate from the low Gibbs free energy zone to the high Gibbs free energy zone, therefore ion transport direction was opposite to the temperature gradient. Simplified finite element simulations based on the coupled PNP and Einstein-Stokes equations were also conducted to verify the influence of the temperature field on directional ion diffusion[50,51]. The simulation model and boundary conditions were described in detail in Supplementary Fig. 43. Our simulation results showed that when the temperature in the nanochannel and reservoirs were consistent, the population and distribution of cations and anions were significantly different in the negative nanochannels due to electrostatic interactions, leading to a significant concentration difference at the junction of the reservoir and channels. When Reservoir-I was heated to 298 K while the nanochannel remained at its initial temperature of 278 K (Fig. 4e, left), a considerable reduction in K$^+$ ion concentration was observed, whereas Cl$^-$ ions exhibited a slight accumulation at the channel orifice. This decrease in K$^+$ ion concentration could be attributed to the initial concentration gradient, which facilitated the diffusion of K$^+$ ions from the nanochannel to the reservoir (Fig. 4e, right). Conversely, the outflow of Cl$^-$ ions driven by the temperature gradient was counteracted by the inflow prompted by the

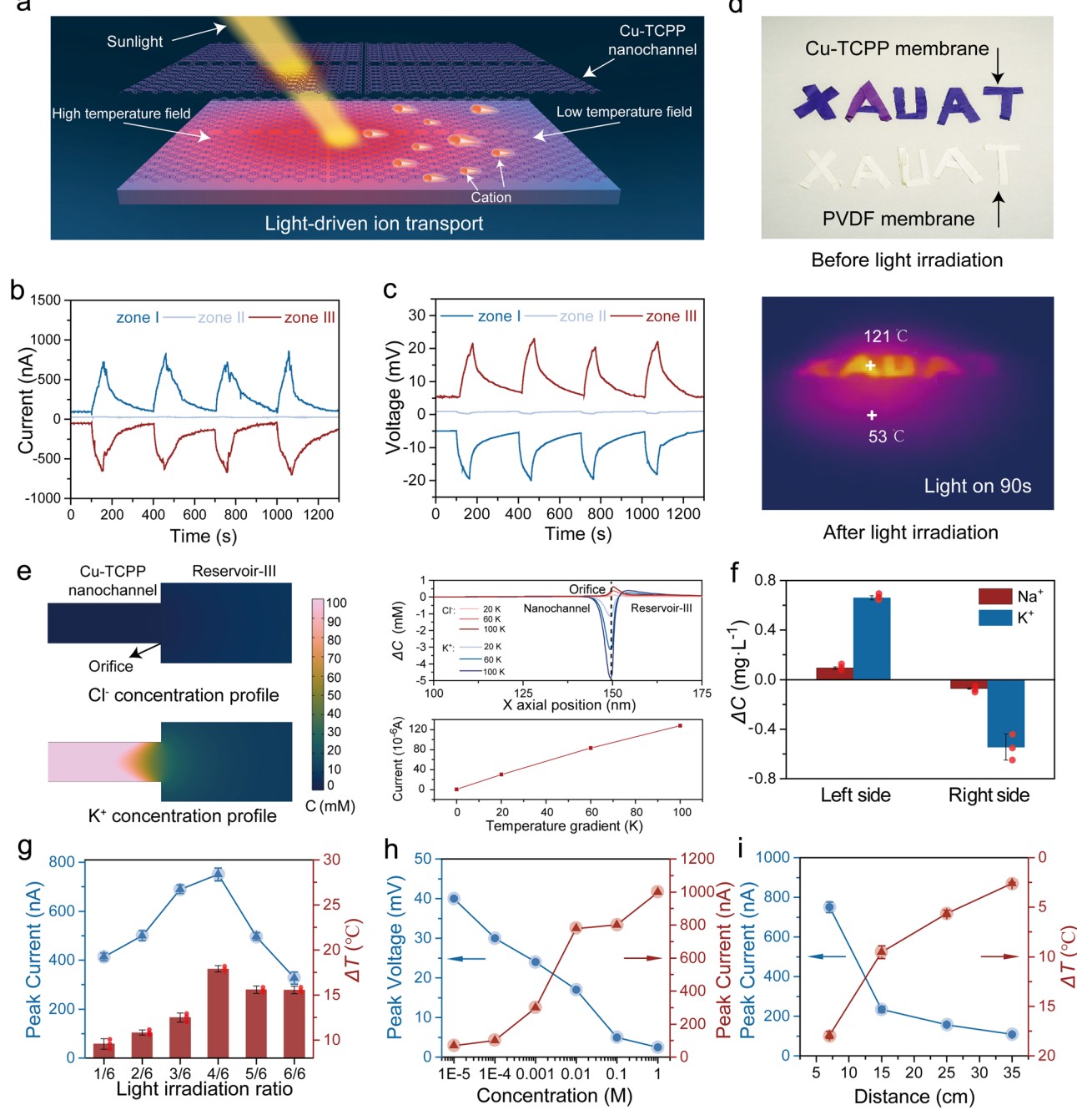

**Fig. 4 | Light-responsive ion transport in Cu-TCPP membranes. a** Schematic diagram of temperature field formed in Cu-TCPP nanochannels under light irradiation. **b, c** Photo-responsive currents and voltages when the light was irradiated on Zone-I, II, and III. **d** Infrared thermal images of Cu-TCPP and PVDF membranes before and after light irradiation. **e** Schematic diagram of the concentration simulation of Cl⁻ (Top) and K⁺ (Down) between the nanochannel and the reservoir (Left). The net change in concentration and ionic current at different temperature gradients (Right). **f** Changes of ion concentration after light-induced transport. Individual data points are plotted to provide a transparent view of the data spread. The error bars represent the standard deviations of three parallel tests. The red point is individual data point which is plotted to provide a transparent view of the data spread. **g** Maximum photocurrent and corresponding temperature difference when different part of Cu-TCPP membrane was illuminated. Individual data points are plotted to provide a transparent view of the data spread. The error bars represent the standard deviations of three parallel tests. The red point is individual data point which is plotted to provide a transparent view of the data spread. **h** Effect of solution concentration on peak voltage and current under light irradiation. **i** Maximum photocurrent and temperature changes under light irradiation with different intensity. The error bars represent the standard deviations of three parallel tests.

concentration gradient, resulting in a minor accumulation of Cl⁻ ions at the orifice. The results also indicated that with an increasing temperature gradient, K⁺ depletion at the orifice became more pronounced, while Cl⁻ ions continued to accumulate (Supplementary Fig. 44)[52]. In addition, based on calculation, as the temperature

gradient increased, the ionic current also increased significantly (Fig. 4e, right).

To provide further experimental evidence for photo-induced active ion transport, the NaCl and KCl solutions with identical concentrations were filled into Reservoir-I and Reservoir-III. We then

measured the ion concentration changes in the two reservoirs before and after illumination using ICP. Due to the photo-induced directed diffusion of $K^+$, the concentration of $K^+$ decreased in Reservoir-III and increased in Reservoir-I after illumination. However, the $Na^+$ concentration in both reservoirs remained almost constant before and after photo-illumination (Fig. 4f and Supplementary Figs. 45 and 46). Moreover, based on the ionic current after filling Reservoir-I and Reservoir-III with 60 °C and 25 °C KCl solutions, respectively, we further confirmed that the direction of the ionic current was consistent with the direction of the low-temperature zone toward the high-temperature zone (Supplementary Fig. 47). Specifically, when the two reservoirs were without a temperature difference, the current value was almost zero. However, when the high-temperature solution was filled in Reservoir-I, a clear ionic current was observed in the same direction as that of the ionic current when Reservoir-I was illuminated.

Various photo-illumination conditions were investigated to optimize the photothermal-electric conversion efficiency of the lamellar Cu-TCPP membrane. To examine the relationship between the illuminated area and photocurrent, the Cu-TCPP membrane was partitioned into six equally sized areas, with non-illuminated sections masked to avoid any light interference. Under conditions where the illuminated area comprised only 1/6 of the total area, the temperature difference ($\Delta T$) between the illuminated and non-illuminated zones was 9.5 °C (Fig. 4h and Supplementary Fig. 48), and only a small number of ions were stimulated to diffuse through the nanochannel. As the illuminated area increased, the temperature difference also increased, resulting in a gradual rise in the photocurrent. At an area ratio of 4/6, the current peaked at 750 nA, with $\Delta T = 17.9$ °C. However, as the illuminated area continued to expand, $\Delta T$ declined, leading to a decay in the photo-responsive current.

In addition, the photocurrent exhibited a positive correlation with increasing concentration, owing to an increase in the number of ion carriers (Fig. 4h and Supplementary Figs. 49 and 50). However, the photovoltage showed a decrease as the concentration continuously increased, which is consistent with previous reports[27]. Moreover, it is worth noting that the photo-responsive currents and voltages were directly proportional to the light intensity (Fig. 4i and Supplementary Fig. 51)[27]. Similarly, photo-responsive ion transport was also observed in other electrolyte solutions (Supplementary Figs. 52 and 53). Additionally, when the continuous light exposure time was extended to 240 s, the net ion current reached a stable 780 nA within 100 s and stabilized around 800 nA, indicating the stability of the photo-induced active ion transport (Supplementary Fig. 54).

## Light-assisted ionic energy conversion

The photothermal-driven ion directional transport phenomenon introduced a promising approach to further enhance the osmotic energy conversion performance of Cu-TCPP-based nanofluidic systems. Here, we first introduced light irradiation into an artificial seawater/river water system. In the absence of light irradiation, a stable current arising from the salinity gradient was observed; while when the low-concentration side was illuminated, the photo-driven ion transport coincided with concentration-driven ion transport, resulting in a noticeable increase in the overall transmembrane ionic currents. Subsequently, upon turning off the light, the ionic photocurrent gradually diminished. The reversible response of ion currents with photo switching suggested the stability and repeatability of light-induced ion transport in this system (Fig. 5a).

When connected to an external resistor, the Cu-TCPP membrane also exhibited a substantial increase in output power with light assistance. The power density in the artificial seawater/river water system increased from the initial 16.64 W m$^{-2}$ without light irradiation to 31.9 W m$^{-2}$ when illuminated at the low-concentration side (Fig. 5b and Supplementary Fig. 55). As described above, the Cu-TCPP membrane already demonstrated superior osmotic power

generation efficiency under normal conditions, owing to its excellent ion selectivity and permeability. These findings further underscored the remarkable potential of the Cu-TCPP membrane to achieve significantly higher power density with light irradiation. Furthermore, the output power density of the Cu-TCPP membrane did not exhibit an obvious decline after 7 days (Fig. 5c), indicating good long-term stability. In addition, the power density exhibited a similar increase as the salinity gradient changed from 5 to 500 times with the help of light (Fig. 5d and Supplementary Fig. 56). At a 500-fold gradient, the power density even reached an impressive 89.65 W m$^{-2}$ under light illumination.

Although researchers have estimated a substantial 1.0 TW of osmotic energy released globally from the mixing of river water and seawater through theoretical calculations[53], the practical utilization of osmotic energy presents significant challenges due to the requirement for freshwater or low-concentration solutions, because both of the two prevailing technologies for osmotic energy harvesting, RED technology, and Pressure Retarded Osmosis (PRO) technology, require solutions of different concentrations on either side of the membrane. In this study, we also investigated whether the conversion of ionic energy into electricity could be realized in normal electrolyte solutions without salinity gradients under light irradiation. In the absence of light irradiation, a linear relationship passing through the origin was observed in the I–V curve when both reservoirs were filled with $10^{-2}$ M KCl solutions (Fig. 5e). Upon irradiation of Reservoir-I, the I–V curve no longer passed through the origin, and a $V_{os}$ and $I_{sc}$ were confirmed by the intercept point, indicating the potential for photothermal-assisted electrical energy conversion in equilibrium electrolyte solutions. Furthermore, a maximum ion power density value of 0.82 W m$^{-2}$ was obtained by connecting an external resistor with the help of photothermal-promoted ion transport (Fig. 5f). Further studies confirmed that light-assisted energy harvesting could be achieved with KCl solutions of varying concentrations (Supplementary Fig. 57). In a pioneering study by Guo et al., the strategy of recovering ionic energy in equilibrium electrolyte solutions was proposed based on 2D-WS$_2$/MoS$_2$ multi-layer heterogeneous membranes[27]. Our group recently confirm the possibility of converting the ionic energy into electricity without salinity gradient using Ti$_3$C$_2$T$_x$/g-C$_3$N$_4$ heterogenous nanochannel with a photo-electric/thermal synergistic effect[51]. The Cu-TCPP membrane in our study demonstrated superior energy conversion efficiency compared to reported values due to its unique structural characteristics and remarkable photothermal effect. Furthermore, light-assisted ionic energy recovery was confirmed in various electrolyte solutions, and the monovalent ions solutions produced more power than divalent ions solutions. (Fig. 5g and Supplementary Fig. 58). The feasibility of ionic energy conversion in the system without salinity gradient presents new possibilities for the recovery of ionic and light energy from diverse saline resources, such as natural seawater conditions, industrial wastewater, and high-saline lake water, and the concept of "salinity energy" is extended to the innovative notion of "ionic energy".

The ions in the Cu-TCPP nanochannel would undergo transport in the same direction when the temperature field induced by asymmetric illumination was opposite to the concentration field. The total current was the sum of the photo-responsive current and the current resulting from a salinity difference (Supplementary Fig. 59). In contrast, when the temperature field direction, induced by asymmetric illumination, aligned with the concentration field, anti-gradient ion transport in the Cu-TCPP membrane could also be achieved. When the ion concentration on the high-concentration side of the solution increased further and the inverse concentration gradient reached 1/50, the recorded total current could not be reversed because the ion current caused by photo-driven ion diffusion was not sufficient to counteract that due to ion diffusion at such a high-concentration gradient (Fig. 5h).

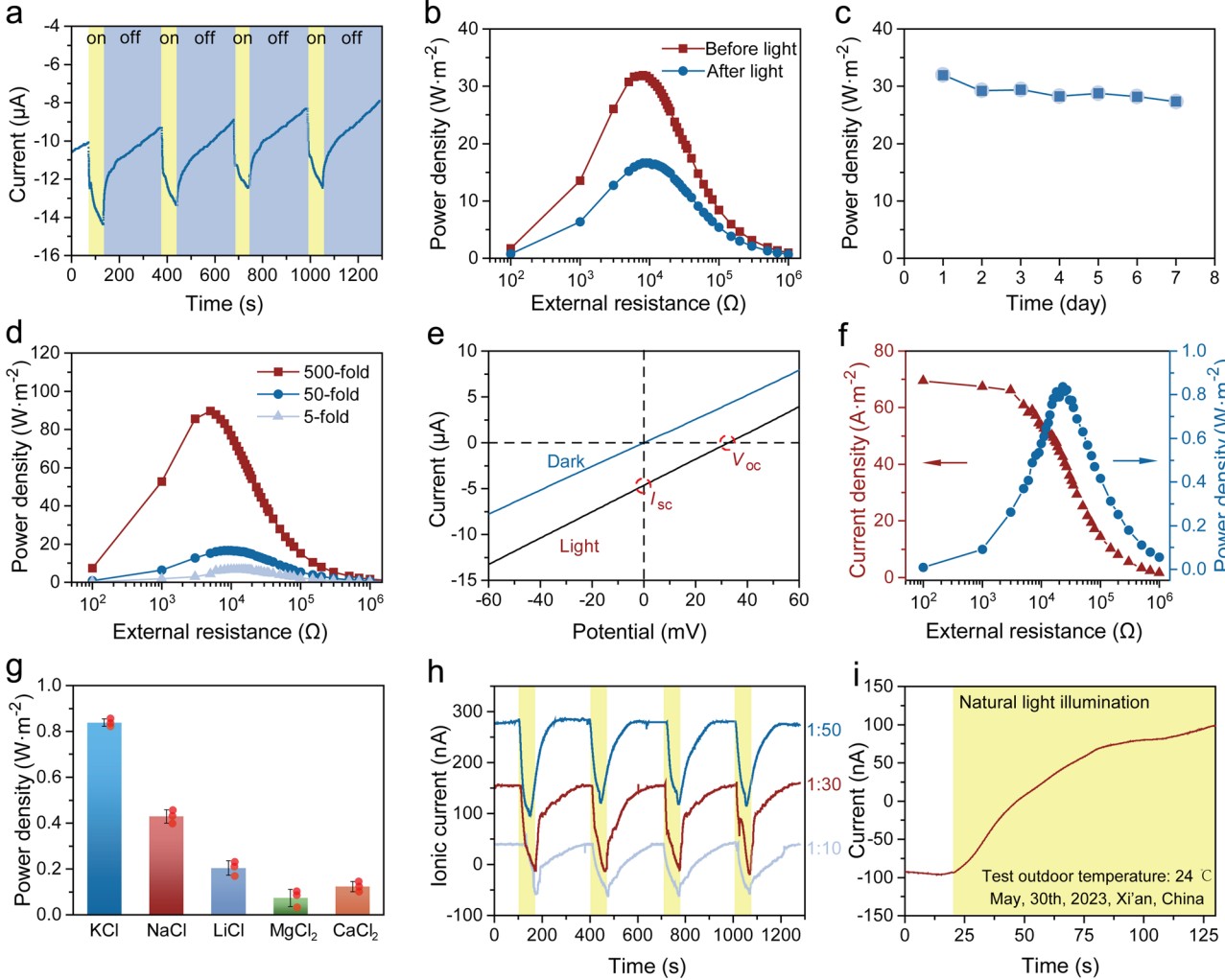

**Fig. 5 | Improved ionic energy conversion performance of Cu-TCPP membrane with light assistance. a** Ionic currents of Cu-TCPP membrane at artificial seawater/river water system before and after light irradiation. The yellow-shaded regions represent the case when illuminated, the blue shaded regions represent uni-lluminated. **b** Power density of Cu-TCPP membrane before and after light irradiation. **c** Long-term stability of energy conversion with light irradiation. **d** Power density of Cu-TCPP membrane tested in KCl solution at different concentration gradients under light irradiation. **e** I–V curves produced under illumination and non-illumination in 0.01 M KCl solution. **f** Power density from equilibrium electrolyte solutions system (0.01 M KCl). **g** Ionic energy conversion performance for various salt solutions without concentration gradient. Individual data points are plotted to provide a transparent view of the data spread. The error bars represent the standard deviations of three parallel tests. The red point is individual data point which is plotted to provide a transparent view of the data spread. **h** Anti-gradient ion transport behavior under a different concentration gradient. The KCl solutions with different concentrations ranging from $2 \times 10^{-4}$-$10^{-3}$ M were used in Reservoir-I and 0.01 M in Reservoir-III. The yellow-shaded regions represent the case when illuminated. **i** Anti-gradient ion transport behavior driven by natural sunlight. The yellow-shaded regions represent the case when illuminated.

The Cu-TCPP-based nanofluidic device was also placed outdoors in a natural environment. When Zone-III was exposed to sunlight under a 20-fold salinity gradient, we observed an increase in ion photo-current from −90 nA to 98 nA within a 100-s, successfully achieving ion transport against the salinity gradient (Fig. 5i and Supplementary Movie 2). These results verify the feasibility of the light-responsive ion transport in Cu-TCPP membrane under natural light conditions.

In our study, we proposed the construction of 2D nanochannel membranes using MOF nanosheets as building units. Recent studies have introduced effective strategies to enhance the osmotic energy conversion efficiency of traditional 2D lamellar membranes. For instance, integrating 1D materials into 2D lamellar membranes significantly boosts power density while enhancing mechanical strength[54]. Moreover, combining materials with different maximum absorption wavelength ranges can further improve light-controlled ion transport efficiency[55]. These innovative approaches provide valuable inspiration for further enhancing the osmotic energy conversion

performance of MOF membranes, thereby propelling osmotic energy harvesting from the realm of laboratory research to practical industrial applications.

## Discussion

In summary, we constructed a nanofluidic system with ultrahigh permeability for light-controlled ion transport by utilizing 2D Cu-TCPP nanosheets. The inherent porous structures of Cu-TCPP and abundant interlayer 2D nanochannels provided rich nanoscale diffusion pathways for selective ion transport. The power density of the osmotic energy conversion device reaches 16.64 W m⁻², surpassing the state-of-the-art nanochannel membranes. When multiple devices were connected in series, the calculator and LED light can function properly. Furthermore, leveraging the excellent photothermal conversion capability of Cu-TCPP, efficient photo-triggered ion active transport was realized. On this basis, the combination of solar energy and salinity gradient significantly enhanced the driving force for ion transport,

further improving the energy conversion performance. Note that, with the assistance of light, the salinity gradient was even no longer a pre-requisite for ion energy harvesting, and the power density in a balanced ion solution system far exceeded that reported in other studies, greatly expanding the application range of ion energy harvesting. Our work shows a novel route of energy conversion and storage from solar energy and the osmotic energy widely existed in seawater, salt-lake brine, and industrial wastewater.

# Experimental section
## Methods
**Material characterization.** The crystallinity and orientation of Cu-TCPP membranes were determined by wide-angle X-ray scattering (WAXS, Xenocs XEUSS, Germany). Microstructures and elemental distribution analysis of Cu-TCPP membrane and nanosheets were achieved through a scanning electron microscope (SEM, Zeiss Gemini SEM 300, Germany). The Cu-TCPP nanosheets and membranes surface images were captured via atomic force microscopy (AFM, Bruker Multimode 8) under tapping mode. The morphology of the nanosheets was characterized by transmission electron microscopy (TEM Talos F200X, USA). The UV-vis measurements of the Cu-TCPP and Porphyrin solutions were performed with an ultraviolet spectrophotometer (UV-3600, Japan). The chemistry property of Cu-TCPP was characterized through X-ray photoelectron spectroscopy (XPS) using a Thermo Scientific ESCALAB Xi⁺ instrument with monochromated Al-Kα radiation. All the binding energies were referenced to the C 1 s peak at 284.8 eV. The FT-IR spectra were recorded using a Bruker VERTEX 33 unit. The zeta potential measurements were measured by dynamic light-scattering analysis using Zetasizer Nano ZS 90. The photothermal performance of Cu-TCPP membranes was monitored by using an Infrared camera (HIKMICRO, China). The ion concentration was determined by an inductively coupled plasma optical emission spectrometer (Focused Photonics, ICP-5000, China).

## Synthesis of Cu-TCPP nanosheets
Bulk Cu-TCPP crystals were successfully synthesized using an optimized thermal solvent method. To ensure complete dissolution of the central copper ion, we sonicated a mixture of $Cu(NO_3)_2 \cdot 3H_2O$, pyrazine in the mixture of N, N-dimethylformamide (DMF), and ethanol for 30 min at room temperature. The ligand solution was prepared by sonication of a solution containing tetra-(4-carboxyphenyl) porphyrin in DMF for 30 min. The metal solution and the ligand solution were mixed and heated in an oil bath at 120 °C for 6 h to produce bulk Cu-TCPP crystals. The resulting purple product was washed once with DMF and twice with ethanol. The dispersion was sonicated in ethanol for one hour, resulting in the exfoliation of the material into monolayer Cu-TCPP nanosheets. The dispersion was subsequently centrifuged ($3500 \times g$, 30 min), and the supernatant containing Cu-TCPP nanosheets was collected for membrane preparation.

## Preparation of 2D Cu-TCPP membrane
We prepared a Cu-TCPP membrane by vacuum filtration of a diluted dispersion of Cu-TCPP nanosheets onto a polyvinylidene fluoride (PVDF) substrate with a pore size of 0.1 μm. The thickness of the resulting Cu-TCPP laminate was controlled by adjusting the volume of the nanosheet dispersion used in the filtration process. After drying in a 60 °C oven overnight, the resulting Cu-TCPP membrane was carefully separated from the substrate. To prepare a large-scale Cu-TCPP membrane with high efficiency, we utilized a custom-designed spray coating device. The process involved pouring the nanosheet dispersion into the reservoir of the spray gun, which was then applied to the surface of the Polyethersulfone (PSE) membrane. The distance between the spray gun and the base membrane was maintained at 20 cm, and the membrane's size was precisely controlled by adjusting the machine program. The spray gun was moved with a speed of

12 m min⁻¹, consistent speed to ensure the homogeneous stacking of the Cu-TCPP layered membrane (Supplementary Fig. 39).

## Device fabrication
The Cu-TCPP membranes were sliced into long strips with dimensions of 30 mm in length, 5 mm in width, and 4 μm in height. The membrane was sealed with a mixture of polydimethylsiloxane (PDMS) prepolymer and curing agent to encapsulate the membrane. The membrane was divided into three equal regions as shown in Fig. 3a.

## Energy conversion measurements
Electrochemical measurements were conducted using a Keithley 2450 sourcemeter with a pair of Ag/AgCl electrodes. The voltage was scanned over a range of −1.5 V to 1.5 V with a step voltage of 0.001 V to obtain the current-voltage (I–V) curve. The open-circuit voltage ($V_{oc}$) and short-circuit current ($I_{sc}$) were determined from the intercepts on the voltage and current axes of the I–V curve, respectively. The number of cation transfers ($t_+$) and the maximum energy conversion efficiency can be calculated using the following equation:

$$(2t_+ - 1) = \frac{E_{diff}}{\frac{RT}{zF} In\left(\frac{\gamma_{cH} C_H}{\gamma_{cL} C_L}\right)} \tag{2}$$

$$\eta_{max} = \frac{1}{2}(2t_+ - 1)^2 \tag{3}$$

Here, $R$, $T$, $z$, $F$, $\gamma$, and $C$ represent the gas constant, temperature, ion valence, Faraday constant, activity coefficient, and salt concentration, respectively. The value of $t_+$ quantifies the selective ion transport across the Cu-TCPP membrane. All permeability tests were conducted at room temperature and repeated a minimum of 3 times to ensure the statistical validity of the results.

## Photothermal effect measurement
A Xenon lamp (Perfect Light Technology, CHF300W) served as a light source for our experimental setup. The photothermal characteristics of the PVDF and Cu-TCPP membranes were monitored using infrared thermography with 2 min irradiation. To simulate the light-induced ion transport, the KCl solution at a temperature close to 60 °C was added into the reservoir, while an equal volume of room temperature KCl solution was added to another reservoir. The ionic current was monitored until the liquid in the reservoirs reached thermal equilibrium.

## Simulation of temperature-responsive ion transport
Numerical simulation was performed using a commercial finite element software package COMSOL (version 4.3) Multiphysics. A numerical simulation based on continuous Poisson-Nernst-Planck (PNP) equations and Einstein-Stokes theory was performed to investigate ion transport behavior in nanochannels under a temperature gradient, as depicted below:

$$\nabla^2 \varphi = -\frac{\rho}{\varepsilon} = -\frac{F}{\varepsilon} \sum z_i c_i \tag{4}$$

$$j_i = -D_i \left(\nabla c_i + \frac{z_i F c_i}{RT} \nabla \varphi\right) \tag{5}$$

$$\nabla \cdot j_i = 0 \tag{6}$$

$$D_i = \frac{k_B T}{6\pi \eta r} \tag{7}$$

Here, the variables $\varphi$, $\rho$, i, $j_i$, $D_i$, $z_i$, and ci represent the electric potential, space charge density, ion species, local ion flux, diffusion coefficient, valence, and concentration of ion species i, respectively. The diffusion coefficients of cation and anion are denoted by $1.957 \times 10^{-9}\,m^2\,s^{-1}$ and $2.032 \times 10^{-9}\,m^2\,s^{-1}$, respectively. Other parameters include the absolute temperature $T$, Faraday constant $F$, gas constant $R$, Boltzmann constant $k_B$, and water constant $\varepsilon$. The dielectric constant is set to 80. Equation (4) is the Poisson equation that describes the relationship between the electric potential and ion concentration. When the system reaches a steady state, the ion flux conforms the time-independent continuity Eq. (6).

## Data availability
The data supporting the findings of this study are available in the paper and its Supplementary Information files. Source data are provided with this paper.

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

## Acknowledgements

This work was supported by National Key R&D Program of China (Nos. 2022YFC2904302), the Key R&D Program of Shaanxi Province (Nos. 2022SF–134), and Education Department of Shaanxi Research Plan (Nos.21JK0709). We also would like to extend gratitude to Prof. Xu Gang for his invaluable assistance in MOF structure analysis

## Author contributions

J.W. and L.W. designed the experiments. J.W., Z.Y.S., X.Y.K., M.L.H., Y.C.Q., S.Z.L., Z.C. and D.W., performed the membrane fabrication and characterization experiments. J.W. and Z.Y.S. wrote the paper, B.H., J.M.H., Z.Y.S. and Y.F. contributed to the project discussions and manuscript writing. J.W., X.Y.K. and L.W. checked the paper.

## Competing interests

The authors declare no competing interests.
