## [Peer Review File · Nature Communications]

Light-Responsive and Ultrapermeable Two-Dimensional Metal-organic Framework Membrane for Efficient Ionic Energy HarvestingREVIEWER COMMENTS

Reviewer #1 (Remarks to the Author):

In this study, Wang et al. have designed a light-responsive nanochannel membrane using a 2D porphyrin-based MOF and investigated its potential application in osmotic energy conversion. The interlayer channels between adjacent MOF nanosheets and intrinsic nanopores within the MOF plane offer abundant pathways for ion diffusion, resulting in a highly permeable membrane with low internal resistance. The authors have also demonstrated the light-thermal conversion capability of the MOF membrane and explored the possibility of eliminating the salinity difference, a requirement in traditional osmotic energy harvesting systems. The work is well-organized and presents interesting and robust results. I believe that its subject matter and quality justify publication in Nature Communications. However, some discussions should be reinforced to enhance clarity and reliability.

1. As a crucial argument, the author is advised to provide additional evidence for faster diffusion in the lateral transport model through additional experiments or numerical simulations.
2. The author provided the concentration difference at the orifice of nanochannel, the changes of ionic current should also be discussed for direct support.
3. The author should conduct a more in-depth literature review on 2D MOF membranes in the introduction, as recent study have used 2D MOF nanosheets as building blocks to prepare membranes (for instance, *Sci. Adv.* 2020; 6: eaay3998).
4. Critical information, such as the ligand-metal reaction, should be added to Fig. 1.
5. Ensure that the wavelength range of the UV-vis test remains consistent to maintain the reliability of statements such as "Pyrazine was not found in the nanosheet dispersion."
6. Deeper analysis of the thickness differences between the reported 2D porphyrin-based MOF nanosheets in various studies should be conducted
7. Revise the statement about surface charge property in relation to Supplementary Fig. 8 to accurately reflect the findings. "This surface charge property remains almost unchanged in the presence of various electrolyte solutions" appears to be inaccurate.
8. Consider revising the placement of the XPS part to align it with the FTIR discussion for a more cohesive chemical property analysis, the current structure is odd.
9. The title of Fig. 3e should be consistent with the description "but also outperformed state-of-the-art 2D lamellar membranes as shown in Fig. 3e" to avoid misunderstanding.

Reviewer #2 (Remarks to the Author):

This work investigated the formation of membrane materials through the stacking of two-dimensional MOFs (Metal-Organic Frameworks) and their application in salinity gradient energy conversion. By utilizing the stacked space between MOF membrane layers, a power density of 16.64W/m² was

achieved. Furthermore, the authors reported that external illumination could further enhance the power density of salinity gradient energy conversion. While the power density in this work is higher than that of traditional two-dimensional material membranes, it does not exhibit a clear advantage in salinity gradient energy conversion compared to previous research on COF (Covalent Organic Framework) materials.

Previous studies have reported that COF membranes with a thickness of 70nm can achieve a power density of 53W/m² (Nature Communication 2022, 13, 3935) which is significantly higher than the results obtained in this work. Additionally, MOFs themselves are known for their high intrinsic pore density, raising questions about the choice of studying stacking channels for salinity gradient energy conversion in this work. Moreover, the results of external illumination improving power performance in this work has been previously reported and does not introduce new physical or chemical concepts.

I believe that this work does not address the critical issues in this field, such as practical concerns like interfacial resistance at a large scale. Furthermore, it does not exhibit advantages in material systems and device structures compared to existing research. Therefore, I do not recommend the publication of this work in Nature Communications.

Reviewer #3 (Remarks to the Author):

The authors proposed a bionic nanofluidic membranes with exceptional high ions permeability and power-production ability even under the natural sunlight and symmetric solutions system. Although the membrane performance, in cases of solar/ionic energy conversion, seems impressive, this manuscript could not be published in Nature Communication at the current stage due to the following reasons.

1. Regarding the conclusion demonstrated in Intro Part, "horizontal ion transport channels outperformed traditional vertical ion transport channels in 2D nanochannel membranes", this so-called conclusion is strange for this reviewer. There is no more description on how this conclusion was gained.
2. Other researches also reported the MOF membranes for energy conversion. The authors only assigned the difference to the enhanced completely utilized interlayer nanochannels for ions transport. This reviewer thinks with conservative gratitude that this alone is not sufficient. From the novelty of perspective, the authors are recommended to initiate some more points to strengthen the superiority and uniqueness of this research. Besides, this reviewer is concerned the true differentiations and superiorities of interlayer transport channels when compared to that of MOF membranes which may have its own intra -/inter crystalline defects for ions high-speed transportation.
3. From the AFM and XRD results, how could the conclusion of "the ions exhibited a tendency to diffuse within the interlayer spacing between the few-layer Cu-TCPP nanosheets rather than between the monolayer Cu-TCPP". The relevant interpretation is rather ambiguous and confusing.
4. This reviewer is confused about the statements of high ions selectivity of the as-performed bionic membranes. The high ions selectivity performance was not eye-catching in this manuscript. Therefore, the reader could not figure out the target ions pairs which might be separated by the stimulus-response Cu-TCPP nanosheet membrane. The underlying mechanism of the ions separation mechanisms should be strengthened. According to the limited knowledge of this reviewer and the presented information in

this manuscript, the interlayer space of the as formed Cu-TCPP was 0.45 nm, and the intrinsic pore of the Cu-TCPP membrane was 0.84 nm. Both of the slit pore and the in-plane pore were incapable of separating ions. In addition, there are much more 2D MOF COF HOF materials, why the authors chose the Cu-TCPP materials as the main research target, how about other porous materials when compared to that of Cu-TCPP.

5. The true seawater and river water other than the model seawater are recommended. Truly, the NaCl was not the sole component in the real circumstance. Besides, the economic-technologic analysis are strongly recommend to be added in this manuscript.

6. It would be better for the authors to show the relationship between the optimized membrane thickness and the best ions transport/energy conversion properties.

7. About the structural stability of the as-formed membranes, some simple dipping and sonication experiments were not so convincing. Even if the authors stated much more advantages and superiorities of this work, the practical significance of the research work and the huge economic/energy effects it may bring in the future should be assessed and evaluated.

Review #1

General comment

In this study, Wang et al. have designed a light-responsive nanochannel membrane using a 2D porphyrin-based MOF and investigated its potential application in osmotic energy conversion. The interlayer channels between adjacent MOF nanosheets and intrinsic nanopores within the MOF plane offer abundant pathways for ion diffusion, resulting in a highly permeable membrane with low internal resistance. The authors have also demonstrated the light-thermal conversion capability of the MOF membrane and explored the possibility of eliminating the salinity difference, a requirement in traditional osmotic energy harvesting systems. The work is well-organized and presents interesting and robust results. I believe that its subject matter and quality justify publication in Nature Communications. However, some discussions should be reinforced to enhance clarity and reliability.

Response to the General Comment

We thank the reviewer very much for the positive evaluation of our work. All the revised text can be found in the manuscript with changes marked.

Comment 1

As a crucial argument, the author is advised to provide additional evidence for faster diffusion in the lateral transport model through additional experiments or numerical simulations.

Response to Comment 1

We thank the reviewer for the kind comment. We first conducted experimental investigations to substantiate the advantages of the horizontal ion transport model over the vertical model. Under nearly identical salt gradients and a similar test area, despite the membrane length in the horizontal model (3 cm) being significantly longer than the membrane thickness in the vertical model (1 μm), the current in the horizontal model was closed to that of the vertical model (Supplementary Figs. 24 and 25), indicating lower resistance for ion transmembrane transport in the horizontal model.

Furthermore, we also employed the Poisson-Nernst-Planck (PNP) equations to

compute net ion currents across the cross-section for both transport modes to validate that the horizontal model is indeed more effective for ion transport than the vertical model. A comprehensive set of parameters for both models is provided in Supplementary Fig. 26 and Table 3. As depicted in Supplementary Fig. 27, when the concentration gradient of the nanofluidic system remains constant on both sides of the reservoir, an elevation in temperature on the reservoir-III side resulted in a considerably higher net current in the horizontal model in comparison to the vertical model. Similarly, an increase in concentration on the reservoir-I side also led to a higher ionic current produced by in the horizontal model compared to the vertical model.

Moreover, for the study on the light-responsive ion transport, the horizontal model also provided the possibility to amplify the temperature gradient within the nanochannel, creating a more significant temperature difference to enhance the impact on ion transport efficiency. To validate this mechanism, we monitored temperature changes in Cu-TCPP membranes under both horizontal and vertical transport modes in the presence of light. In the horizontal model, one end of the membrane was illuminated, and the temperature difference by comparing the illuminated section with the opposite end was measured. In vertical model, we illuminated the front surface of the membrane and measured the temperature difference between the illuminated side and the opposite side of the membrane (Supplementary Fig. 39). The temperature difference in the Cu-TCPP membrane increased to 120°C in just 90 seconds in the horizontal transportation model, whereas in the vertical transportation model, under the same irradiation conditions, the temperature difference was only 5°C. A detailed description has been added in the revised manuscript as follows:

“Experimental investigations were conducted to validate that the horizontal model is indeed more effective for ion transport than the vertical model. Under nearly identical salt gradients and a similar test area, even though the membrane length in the horizontal model (3 cm) was significantly longer than the membrane thickness in the vertical model (1 μm), the current in the horizontal model nearly matched that of the vertical model, indicating lower resistance in ion transmembrane transport in the

horizontal model (Supplementary Figs. 24 and 25). Furthermore, we employed the Poisson-Nernst-Planck (PNP) equations to compute net ion currents across the cross-section under salinity gradient for both transport modes. A comprehensive set of parameters for both models were provided in Supplementary Figs. 26, 27, and Table 3. An increase in concentration on the reservoir-I side led to a higher net current produced by the horizontal model compared to the vertical model.

In addition to the lower transport resistance in the vertical model, the horizontal model also provided the possibility to amplify the temperature gradient within the nanochannel, creating a more significant temperature difference to enhance the impact on ion transport efficiency. To validate this mechanism, we monitored temperature changes in Cu-TCPP membranes under both horizontal and vertical transport modes in the presence of light. In the horizontal model, one end of the membrane was illuminated, and the temperature difference by comparing the illuminated section with the opposite end was measured. In vertical model, we illuminated the front surface of the membrane and measured the temperature difference between the illuminated side and the opposite side of the membrane (Supplementary Fig. 39). The temperature difference in the Cu-TCPP membrane increased to 120°C in just 90 seconds in the horizontal transportation model, whereas in the vertical transportation model, under the same irradiation conditions, the temperature difference was only 5°C. Similarly, the PNP equation was utilized to compute the net ion currents in the cross-section for both transport modes under temperature gradient (Supplementary Fig. 27). When the concentration gradient of the nanofluidic system remained constant on both sides of the reservoir, an elevation in temperature on the reservoir-III side resulted in a considerably higher net current for the horizontal model than for the vertical model.”

Supplementary Figure 24. Schematic illustration two different transport models (a) and the membrane size for ionic current comparison (b).

Supplementary Figure 25. Stabilized ionic currents in the horizontal and vertical transport model.

Supplementary Figure 26. Schematic illustration of vertical transportation model (a), horizontal transportation model (b).

Supplementary Table 3. List of parameters for numerical simulation model.

Parameter	Description	Value
A	Length of reservoir	100 nm
B	Width of reservoir	100 nm
D	Diameter of nanochannel	10 nm
L	Length of nanochannel	25 nm
σ	Surface charge density	-0.06 C/m ²
$d1$	Interlayer spacing	2 nm
$d2$	Width of nanosheet	0.5 nm
D_p	Diffusion coefficient of K ⁺	1.957×10^{-9} m ² /s
D_n	Diffusion coefficient of Cl ⁻	2.032×10^{-9} m ² /s
$c0$	Electrolyte concentration	1-50 mM
T	Temperature	298-378 K

Supplementary Figure 27. The current difference at the orifice of the horizontal and vertical transport model under different salinity gradients/temperature gradients. X-axis represented the temperature of reservoir-III; y-axis represented the concentration of reservoir-I; and z-axis represented the current difference.

Supplementary Figure 39. Schematic of the temperature measurement for two models for under illumination. (a) In the horizontal model, the temperatures of illuminated and unilluminated ends were measured by infrared thermographic camera. (b) In the vertical model, the temperature of illuminated surface was measured by infrared thermographic camera, and the opposite side of the membrane was monitored by connecting to a thermocouple.

Comment 2

The author provided the concentration difference at the orifice of nanochannel, the changes of ionic current should also be discussed for direct support.

Response to Comment 2

We thank the reviewer for the kind comment. Following the reviewers' suggestion, we performed theoretical calculations of ionic currents under various temperature gradients (ion concentration: 10 mM; temperature differences: 0K, 20K, 60K, and 100K) based on PNP equation. It was observed that as the temperature difference increased gradually, the driving force for ion transport was reinforced, resulting in a corresponding increase in the ionic current. A detailed description has been added in the revised manuscript as follows:

Fig. 4 Light-responsive ion transport in Cu-TCPP membranes. **a** Schematic diagram of temperature field formed in Cu-TCPP nanochannels under light irradiation. **b**, **c** Photo-responsive currents and voltages when the light was irradiated on Zone-I, II, and III. **d** Infrared thermal images of Cu-TCPP and PVDF membranes before and after light irradiation. **e** Schematic diagram of the concentration simulation of Cl^- (Top) and K^+ (Down) between the nanochannel and the reservoir (Left). The net change in concentration and ionic current at different temperature gradients (Right). **f** Changes of ion concentration after light-induced transport. **g** Maximum photocurrent and corresponding temperature difference when different part of Cu-TCPP membrane was illuminated. **h** Effect of solution concentration on peak voltage and current under light irradiation. **i** Maximum photocurrent and

temperature changes under light irradiation with different intensity. The error bars represent the standard deviations of three parallel tests.

“The results also indicated that with an increasing temperature gradient, K^+ depletion at the orifice became more pronounced, while Cl^- ions continued to accumulate (Supplementary Fig. 41)⁵⁰. In addition, based on calculation, as the temperature gradient increased, the ionic current also increased significantly (Fig. 4e, right).”

Comment 3

The author should conduct a more in-depth literature review on 2D MOF membranes in the introduction, as recent study have used 2D MOF nanosheets as building blocks to prepare membranes (for instance, Sci. Adv. 2020; 6: eaay3998).

Response to Comment 3

We thank the reviewer for the kind comment. We sincerely apologize for the incomplete literature research on the 2D MOF lamellar membrane. Upon further investigation, we have found that the utilization of 2D MOFs as building blocks for membrane for mass separation application have been reported in several recent studies. However, to the best of our knowledge, the application of 2D MOF lamellar membranes in osmotic energy recovery remains a rarity. The recent studies about 2D MOF membrane have been added to the reference list in the revised manuscript as follows:

~~“The emergence of metal-organic frameworks (MOFs)²¹ and covalent organic frameworks (COFs)²² with intrinsic, regular, and uniformly sized pores has provided new opportunities for enhancing osmotic energy conversion performance. However, the reported MOF membranes for osmotic energy harvesting were mainly constructed by 3D MOF crystals^{23,24}. Achieving optimal osmotic energy conversion efficiency by constructing 2D MOF layer membranes that fully utilize the interlayer nanochannels remains unattainable²⁵. Recent studies have also introduced 2D MOF nanosheets with intrinsic, regular, and uniformly sized pores for constructing MOF layered membranes¹⁷. Leveraging the both the in-plane pores and interlayer transport pathways, these 2D MOF membranes have demonstrated considerable promise in gas~~

and molecular separation¹⁸⁻²¹. However, the utilization of 2D MOF membrane in osmotic energy conversion applications remains notably scarce.”

Reference:

(17) Li X, et al. Construction of angstrom-scale ion channels with versatile pore configurations and sizes by metal-organic frameworks. *Nat. Commun.* **14**, 286 (2023).

(18) Cheng P, et al. Two-dimensional metal-porphyrin framework membranes for efficient molecular sieving. *J. Membrane Sci.* **640**, 119812 (2021).

(19) Zhou Y, et al. Highly-Efficient Ion Gating through Self-Assembled Two-Dimensional Photothermal Metal-Organic Framework Membrane. *Angew. Chem. Int. Ed.* **62**, e202302997 (2023).

(20) Li G, et al. Ni-metal-organic-framework (Ni-MOF) membranes from multiply stacked nanosheets (MSNs) for efficient molecular sieve separation in aqueous and organic solvent. *J. Membrane Sci.* **635**, 119470 (2021).

(21) Song H, et al. Structure Regulation of MOF Nanosheet Membrane for Accurate H₂/CO₂ Separation. *Angew. Chem. Int. Ed.* **62**, e202218472 (2023).

Comment 4

Critical information, such as the ligand-metal reaction, should be added to Fig. 1.

Response to Comment 4

We thank the reviewer for the kind comment. The ligand-metal reactions, including the formation of Cu-TCPP unit and the paddle-wheel metal node, were included in the schematic in the revised manuscript as follows:

Fig. 1 Schematic illustration of the preparation of 2D Cu-TCPP lamellar membrane with excellent photo-thermal effect.

Comment 5

Ensure that the wavelength range of the UV-vis test remains consistent to maintain the reliability of statements such as "Pyrazine was not found in the nanosheet dispersion."

Response to comment 5

We thank the reviewer for the kind comment. We apologize for the mismatched wavelength range in the UV-vis testing of Cu-TCPP nanosheet dispersion and pyrazine. Additional tests were conducted, confirming that, after thorough washing, pyrazine was almost absent in the Cu-TCPP nanosheet dispersion. A detailed description has been added in the revised manuscript as follows:

~~“Pyrazine, which assists the metal ions in binding to the ligand, was not found in the nanosheet dispersion (Supplementary Fig. 2)³³. Pyrazine, which assists the metal ions in binding to the ligand, was almost absent in the Cu-TCPP nanosheet dispersion after thorough washing (Supplementary Fig. 2)²⁹.”~~

~~Supplementary Figure 2. UV-vis spectra of the pyrazine dispersion. UV-Vis spectra of pyrazine and Cu-TCPP dispersion.~~

Comment 6

Deeper analysis of the thickness differences between the reported 2D porphyrin-based MOF nanosheets in various studies should be conducted

Response to Comment 6

We thank the reviewer for the kind comment. Based on a literature review, it is evident that the thicknesses of most reported TCPP-based MOF nanosheets, as measured by AFM, fall within the range of 2-5 nm, as documented in the following

Table R1. A limited number of studies have reported that the thickness of the prepared nanosheets closely approximates the theoretical thickness of 0.27 nm. In our present study, the thickness of the Cu-TCPP nanosheets prepared was approximately 2.3 nm, as determined through AFM measurements.

Table R1. Thickness of reported 2D porphyrin-based MOF nanosheets.

Material	Thickness of nanosheets (nm)	Refs.
PMOF	1.9	1
Al-MOF	1.9	2
Cu-TCPP	2.0	3
Cu-TCPP	4.0	4
Al-TCPP	17±1.2	5
Cu-TCPP	0.8	6
Zn-TCPP	7.6±2.6	
Cu-TCPP	4.5±1.2	
Co-TCPP	18.8±6.4	7
Zn-TCPP(Fe)	7.4±2.9	
Cu-TCPP	5.4	8
MOF	4~5	9
Cu-TCPP	3.0	10
Cu-TCPP	4~5	11
Cu-TCPP	1.5±0.5	12
Co-TCPP	40	13
Cu-TCPP(Fe)	1.0	14
Zn-TCPP(Fe)	1.9±0.1	15
Cu-TCPP	25	16
In-TCPP	3.97	17
M-TCPP	7.7±3.5	18
Cu-TCPP	6.5	
Co-TCPP	4.8	19
Ce-TCPP	3.7	
Cu-TCPP	2.9	20
Pd-TCPP	1.38±0.24	21
MPFs	2.96	22
Cu-TCPP	2.6	23
Cu-TCPP	10	24
Cu-TCPP	5.25	25
Cu-TCPP(Fe)	10	26
Zn-TCPP	2.7	27
Cu-TCPP	20	28
Cu-TCPP	2.5	29
Zn-TCPP	Less than 5	30

Cu-TCPP	0.27	31
Cu-TCPP	2.3	This work

Reference:

(1) Zhang Y, *et al. CCS Chem.* **4**, 3329-3341 (2022); (2) Jian M, *et al. Sci. Adv.* **6**, eaay3998 (2020); (3) Hussain S, *et al. J. Membrane Sci.* **620**, 118888 (2021); (4) Chen J, *et al. J. Membrane Sci.* **648**, 120382 (2022); (5) Xiao Y, *et al. Micropor. Mesopor. Mat.* **324**, 111272 (2021); (6) Cai J, *et al. Nano Res.* **16**, 6290-6297 (2023); (7) Zhao M, *et al. Adv. Mater.* **27**, 7372-7378 (2015); (8) Zhou Y, *et al. Angew. Chem. Int. Ed.* **62**, e202302997 (2023); (9) Zhao X, *et al. Nano Energy* **81**, 105657 (2021); (10) Cheng P, *et al. J. Membrane Sci.* **640**, 119812 (2021); (11) Wu G-D, *et al. Angew. Chem. Int. Ed.* **60**, 9931-9935 (2021); (12) Hussain S, *et al. Sep. Purif. Technol.* **268**, 118688 (2021); (13) Chen L, *et al. Nanoscale* **11**, 11121-11127 (2019); (14) Wang C, *et al. Adv. Funct. Mater.* **30**, 1908804 (2020); (15) Ang H, *et al. ACS Appl. Mater. Interfaces* **9**, 28079-28088 (2017); (16) Zhao S, *et al. Environ. Sci. Pollut. R.* **27**, 39186-39197 (2020); (17) Zhang Z, *et al. Nanoscale* **14**, 7146-7150 (2022); (18) Ding G, *et al. Adv. Funct. Mater.* **29**, 1806637 (2019); (19) Li S, *et al. Mol. Catal.* **496**, 111196 (2020); (20) Liang Y, *et al. Sep. Purif. Technol.* **282**, 120045 (2022); (21) Zhang H, *et al. Chem. Eng. J.* **442**, 136144 (2022); (22) Zhao W, *et al. Small* **15**, 1901351 (2019); (23) Wang Z, *et al. J. Membrane Sci.* **633**, 119397 (2021); (24) Wang C, *et al. Chem.* **835**, 178-185 (2019); (25) Li B, *et al. Theranostics* **8**, 4086-4096 (2018); (26) Chen H, *et al. ACS Appl. Mater. Interfaces* **10**, 24108-24115 (2018); (27) Zhang L, *et al. Angew. Chem. Int. Ed.* **60**, 22664-22671 (2021); (28) Xu G, *et al. J. Am. Chem. Soc.* **134**, 16524-16527 (2012); (29) Jian M, *et al. Chem. Commun.* **53**, 13161-13164 (2017); (30) Li X, *et al. ACS Appl. Mater. Interfaces* **11**, 24618-24626 (2019); (31) Yang D, *et al. Chem. Int. Ed.* **59**, 18954-18959 (2020).

The precision of AFM measurements could be influenced by adsorbed surfactants on both flakes and substrates (*Nat Commun.*, 2014, 5, 4576). For example, in our prior research, the height measured for Ti₃C₂T_x nanosheets, with a theoretical thickness of 0.88 nm, was around 1.2 nm under our experimental conditions (*Energy Environ. Sci.*, 2023, 16, 3146-3157). Nevertheless, even after accounting for potential errors stemming from surface adsorption, it is evident that the thickness of the Cu-TCPP nanosheets we prepared significantly exceeds their theoretical thickness. Consequently, we consider the nanosheets produced in this study to be characterized as few-layered nanosheets rather than truly single-layer nanosheets. A detailed description has been added in the revised manuscript as follows:

~~“According to previous studies, the theoretical thickness of monolayer Cu-TCPP nanosheet is 0.28 nm³⁶, while the average thickness of the prepared Cu-TCPP nanosheet, as determined via atomic force microscopy (AFM), were approximately 2.3 nm (Fig. 2d and Supplementary Fig. 4), indicating that the Cu-TCPP nanosheet~~

~~prepared in our study consist of few layers of monolayer nanosheets. To further analyze the structural characteristics of the obtained few-layer Cu-TCPP nanosheet, we conducted the X-ray diffraction (XRD) tests on the in-plane and out-of-plane structures. The peak in the out-of-plane XRD pattern at $\sim 19.6^\circ$ indicated a d-spacing of 4.5 Å for the two adjacent Cu-TCPP monolayers, consistent with previous literature (Supplementary Figs. 5 and 6)^{27,37}~~

~~Combining with the AFM and XRD results presented above, we infer that ions exhibit a tendency to diffuse within the interlayer spacing between the few-layer Cu-TCPP nanosheets rather than between the monolayers Cu-TCPP. Additionally, the large number of regular nanopores in the MOF nanosheets could provide "shortcuts" for ion transport, which could enhance the ion flux^{40,41}~~

Previous studies have indicated that the theoretical thickness of a monolayer Cu-TCPP nanosheet is 0.27 nm³³. However, the average thickness of the prepared Cu-TCPP nanosheet, as determined via atomic force microscopy (AFM), measured approximately 2.3 nm (Fig. 2d and Supplementary Fig. 4). The precision of AFM measurements could be influenced by adsorbed surfactants on both flakes and substrates³⁴. However, even after accounting for errors caused by surface adsorption, the thickness of the Cu-TCPP nanosheets we prepared far exceeded their theoretical thickness. Hence, we conclude that the nanosheets prepared in this study are multi-layered rather than truly single-layered nanosheets.”

Comment 7

Revise the statement about surface charge property in relation to Supplementary Fig. 8 to accurately reflect the findings. "This surface charge property remains almost unchanged in the presence of various electrolyte solutions" appears to be inaccurate.

Response to Comment 7

We thank the reviewer for the kind comment. We have revised the initial imprecise statement about changes in zeta potential in different electrolyte solutions. A detailed description has been added in the revised manuscript as follows:

~~“This surface charge property almost unchanged in the presence of various electrolyte solutions (Supplementary Fig. 8). As illustrated in Supplementary Fig. 8, nanosheets' inherent negative charge characteristics nearly remained consistent in various electrolyte solutions. In the presence of a CaCl₂ solution, the surface negative charges exhibited a modest decrease, which could be attributed to the pronounced screening effect exerted by the divalent Ca²⁺ ions.”~~

Comment 8

Consider revising the placement of the XPS part to align it with the FTIR discussion for a more cohesive chemical property analysis, the current structure is odd.

Response to Comment 8

We thank the reviewer for the crucial comment. We have incorporated the discussion of XPS results into the chemical property analysis part, enhancing the overall readability and coherence of the nanosheet preparation part. A detailed description has been added in the revised manuscript as follows:

~~“Furthermore, to investigate the chemical structure of Cu-TCPP membranes, X-ray photoelectron spectroscopy (XPS) tests were performed (Supplementary Fig. 23). The C 1s spectrum showed fitted peaks at about 284.8, 286.2, and 288.3 eV, representing the C-C, C-N, and C=O/C-O bonds. In addition, the high-resolution Cu 2p and O 1s spectra showed the presence of Cu₂(COO)₄ nodes. The binding energy peak of 944.3 eV in the Cu spectrum corresponded to the characteristic peak of Cu²⁺; peaks centered at 934.8 and 954.7 eV indicated the coexistence of two Cu valence states (Cu⁺ and Cu²⁺). In addition, the N 1s XPS spectrum demonstrated at 398.6 eV and 400.3 eV, representing N-Cu and N-H peaks, confirming the coordination of the copper ion to the nitrogen atom in the center of the porphyrin ring^{27,42}.~~

The peak at 1700 cm⁻¹ almost disappeared, suggesting the decreasing C=O stretching vibration and the formation of Cu₂(COO)₄ paddlewheel metal nodes^{30,31}. Based on the X-ray photoelectron spectroscopy (XPS) tests results of Cu-TCPP membranes (Supplementary Fig. 3), the C 1s spectrum showed fitted peaks at about 284.8, 286.2, and 288.3 eV, representing the C-C, C-N, and C=O/C-O bonds. In addition, the

high-resolution Cu 2p and O 1s spectra showed the presence of $\text{Cu}_2(\text{COO})_4$ nodes. The binding energy peak of 944.3 eV in the Cu spectrum corresponded to the characteristic peak of Cu^{2+} ; peaks centered at 934.8 and 954.7 eV indicated the coexistence of two Cu valence states (Cu^+ and Cu^{2+}). In addition, the N 1s XPS spectrum demonstrated at 398.6 eV and 400.3 eV, representing N-Cu and N-H peaks, confirming the coordination of the copper ion to the nitrogen atom in the center of the porphyrin ring^{23,32}.”

Review #2

General comment

This work investigated the formation of membrane materials through the stacking of two-dimensional MOFs (Metal-Organic Frameworks) and their application in salinity gradient energy conversion. By utilizing the stacked space between MOF membrane layers, a power density of 16.64W/m^2 was achieved. Furthermore, the authors reported that external illumination could further enhance the power density of salinity gradient energy conversion. While the power density in this work is higher than that of traditional two-dimensional material membranes, it does not exhibit a clear advantage in salinity gradient energy conversion compared to previous research on COF (Covalent Organic Framework) materials.

Previous studies have reported that COF membranes with a thickness of 70nm can achieve a power density of 53W/m^2 (Nature Communication 2022, 13, 3935) which is significantly higher than the results obtained in this work. Additionally, MOFs themselves are known for their high intrinsic pore density, raising questions about the choice of studying stacking channels for salinity gradient energy conversion in this work. Moreover, the results of external illumination improving power performance in this work has been previously reported and does not introduce new physical or chemical concepts.

I believe that this work does not address the critical issues in this field, such as practical concerns like interfacial resistance at a large scale. Furthermore, it does not exhibit advantages in material systems and device structures compared to existing research. Therefore, I do not recommend the publication of this work in Nature Communications.

Response to the General Comment

We really appreciate the reviewer's efforts on our work and thank the reviewer for providing insightful comments to further improve the quality of our manuscript. Point-by-point responses to the comments are enclosed as follows:

1. Performance comparison with previously reported COF membrane

The power density stands as a pivotal parameter for evaluating the efficiency of osmotic energy conversion of nanofluidic membrane. Previous studies have demonstrated that power density correlates not only with the ion selectivity and permeability of the membrane but also with external variables such as effective testing area and solution composition. For instance, empirical and theoretical analyses have shown a substantial decrease in power density with an increase in testing area. Gao et al. reported the power density dropped by more than one order of magnitude as the testing area expanded from 0.03 mm² to 7.07 mm² (*Angew. Chem. Int. Ed.*, 2023, 62, e202218129). Although the power density value of 16.64 W/m² in our investigation is a quarter of the reported power density of 2DPI membranes indicated by the reviewer (*Nat. Commun.*, 2022, 13, 3935), it's crucial to emphasize that the **testing area for their 2DPI membrane was 12 μm², roughly 1600 times smaller than that of our Cu-TCPP membrane.**

Additionally, despite the excellent energy conversion performance exhibited by the 2DPI membrane in experiments, the authors acknowledged a significant challenge in their paper. They stated, "*We would also like to mention that the 2DPI membranes using SMAIS method are upscalable. Nevertheless, there is still a long way to go for large-scale applications because an enormous membrane area will be needed for real reverse electrodialysis pilot-scale setup.*" **This description underscores the difficulty in achieving scalable production of this 2DPI membrane with a thickness of 70 nm using the reported surfactant-monolayer-assisted interfacial synthesis (SMAIS) method.**

In contrast, one of the significant reasons 2D lamellar membranes have garnered substantial attention in the separation field is their potential for industrial-scale production (*Nat. Commun.*, 2016, 7, 10891; *J. Membrane Sci.*, 2020, 612, 118454). In our study, by facilely enlarging the size of the substrate membrane in vacuum filtration method, we successfully produced a Cu-TCPP membrane with an area of 78.5 cm² (Supplementary Fig. 33). Given the low production efficiency and time-consuming of the vacuum filtration method, **we also tried to fabricate a Cu-TCPP membrane with a larger area of 600 cm² at high speed via the spraying**

coating strategy, further highlighting the potential for large-scale production and industrial applications of our 2D Cu-TCPP membrane (Supplementary Fig. 36).

2. Choice of MOF nanosheet restacking model

Owing to their inherent porous structure, MOFs have long been considered a promising material for constructing separation membranes. Especially, recent reported have highlighted the development of various 2D MOF membranes which demonstrated their exceptional performance in applications such as molecular sieving and gas separation (*J. Membrane Sci.* 2021, 635, 119470; *Sci. Adv.*, 2020, eaay3998; *Desalination*, 2023, 567, 116979).

Nevertheless, based on our literature survey, the MOF membranes employed for osmotic energy recovery predominantly originate from 3D MOF crystals (*Nano Energy*, 2018, 53, 643-649; *Sci. Adv.*, 2021, 7, eabe9924; *ACS Nano*, 2022, 16, 16343-16352), and **the application of 2D MOF membranes in osmotic energy recovery is notably absent**. Consequently, in this study, we fabricated a 2D Cu-TCPP MOF layered membrane and explored the potential for enhancing energy conversion performance through its intrinsic pores in the MOF nanosheets and interlayer channels between adjacent nanosheets. If our literature review lacks comprehensiveness in any aspect, we kindly request the reviewers to provide feedback. We are more than willing to make necessary revisions to ensure the completeness and accuracy of our work.

Furthermore, since the precise ion separation ability of interlayer nanochannels were confirmed in 2012, 2D membranes constructed by the parallel stacking of nanosheets have garnered significant research interest due to their combined advantages (*Angew. Chem. Int. Ed.*, 2023, 62, e202218321). For instance, chemical modifications can be conducted in the solution phase before preparing the 2D membrane, markedly enhancing efficiency of channel geometry and chemical properties modification. Additionally, when combined with zero-dimensional (0D) or one-dimensional nanomaterials (1D) to construct heterogeneous structures, the synergistic exploitation of different materials' inherent advantages can further enhance energy conversion performance. Moreover, it is possible to create more sophisticated biomimetic

structures to achieve novel ion transport mechanisms.

Following the initial demonstration of the potential for effective osmotic energy retrieval using 2D MOF membranes in this study, we will endeavor to further explore the full potential of 2D MOF in our future research through chemical modifications, integration with composite materials, or the design of biomimetic structures as described above to enhance energy harvesting efficiency.

3. Differences in light-enhanced energy conversion performance compared to previous reports

According to theoretical calculations, an estimated 1.0 TW of osmotic energy is released globally from the mixing of river water and seawater (*Nature*, 2012, 488, 313-31). However, **a substantial amount of freshwater or low-salinity river water is necessary when considering comprehensive energy harvesting from seawater worldwide** because both of the two prevailing technologies for osmotic energy harvesting, Reverse Electrodialysis (RED) technology and Pressure Retarded Osmosis (PRO) technology, require solutions of different concentrations on either side of the membrane.

To address this challenge, we tried to utilize the exceptional photothermal properties of Cu-TCPP to harness energy in salt solutions with uniform concentrations. When the concentrations of salt solutions on both sides of the Cu-TCPP membrane were equal, there was no traditional salt gradient to drive ion transport. However, under light illumination, a clear ionic current was observed because a temperature gradient formed within the nanochannels. On this basis, **the electrical energy was successfully generated without a conventional salt concentration gradient with the help of light**, breaking the limitation of osmotic energy conversion to salt-gradient systems and significantly reducing practical application challenges.

Currently, only two studies have reported similar energy conversion behavior in equilibrium electrolyte solutions. Guo et al. utilized the formation and migration of photoexcited charge carriers between WS₂/MoS₂ to achieve a power density of 2 mW/m² in a 10⁻⁴ M KCl solution (*Adv. Mater.* 2021, 33, 2007529). Additionally, our team previously prepared a Ti₃C₂T_x/g-C₃N₄ heterogeneous nanochannel membrane,

leveraging photoelectric and photothermal performance synergies, resulting in a power density of 21.93 mW/m² in the same solution environment (*Energy Environ. Sci.*, 2023, 16, 3146-3157). In contrast, owing to the exceptional photothermal properties of Cu-TCPP, our present work achieved a remarkable power density of up to 840 mW/m² in the same equilibrium electrolyte environment. This value is much higher than that reported by Guo et al. and our previous research findings.

In summary, while many studies have explored enhancing osmotic energy conversion efficiency through light illumination, our study takes a step forward. First, the osmotic energy conversion efficiency of the Cu-TCPP membrane was significantly enhanced by light illumination, surpassing the power density of state-of-the-art membranes under similar measurement conditions. More importantly, **our research marks a paradigm shift by demonstrating that electrical energy extraction is no longer limited to traditional salt-gradient systems**. As a result, we have pioneered the concept of “osmotic energy” into “ionic energy”, which holds vital implications for resource utilization applications, encompassing both anthropogenic sources like mining wastewater and seawater desalination brine and natural sources such as salt lakes brine and brackish water.

4. Analysis of interfacial resistance in the RED system

As suggested by the reviewer, in the RED system, interface resistance exerts a great impact on osmotic energy conversion performance. Specifically, an unavoidable concentration polarization phenomenon is observed at the entrance of the nanochannel, causing ion depletion in this region and consequently increasing ion transport resistance (*Angew. Chem. Int. Ed.* 2023, 62, e202218129). The simulation based on the PNP equation indicated that under ideal conditions, without considering the influence of interface resistance, the output power linearly increased with membrane thickness. However, in practical environments, inevitable interfacial resistance could not be ignored, leading to the output power often lower than the ideal value (*Small* 2019, 1804279). As a result, achieving a high power density with a large-scale testing area has been identified as a critical challenge in the practical application of osmotic energy harvesting. It could be noticed that power densities of reported nanochannel

membranes have been typically evaluated within areas smaller than 0.05 mm².

Inspired by this valuable comment, we further investigated the influence of the testing area on the power density of our Cu-TCPP membrane (Supplementary Fig. 29). Similar to prior literature reports, the power density decreased as the testing area of the membrane expanded. However, we could still find that our Cu-TCPP membrane can yield a power density of 4.32 W/m² even with a testing area as large as 0.2 mm² in the artificial seawater and river system. This value not only approached the commercialization benchmark of 5 W/m² but also competitive in the reported power densities of membranes tested within such expansive areas under similar concentration difference systems.

We wholeheartedly agree with the reviewer's comment regarding the crucial issue of interfacial resistance in permeation energy. We apologize for not addressing this matter in our current study. However, enhancing power density for large-area membranes is undoubtedly a primary objective in our future research due to its paramount importance in bridging the gap between experimental studies and practical energy recovery applications.

In summary, by utilizing the inherent porous structure and photothermal conversion property of 2D Cu-TCPP MOF, we successfully prepared a lamellar nanofluidic membrane for highly efficient osmotic energy conversion. The membrane achieved an impressive power density of 16.64 W/m², surpassing that of reported membranes under similar solution systems. Leveraging the excellent photothermal conversion capability of Cu-TCPP could further enhance energy conversion performance. Importantly, light illumination significantly augmented the driving force for ion transport, rendering salinity gradient no longer a prerequisite for ion energy harvesting. With the assistance of light, electricity energy could be generated from common electrolyte solutions. While we have primarily confirmed the large-scale production potential and explored the feasibility of powering practical electronic devices, we recognize the need to focus more on device design, as kindly suggested by the reviewer. This strategic emphasis will enable us to promote the transition from laboratory research to practical implementation. A detailed description has been added

in the revised manuscript as follows:

“Large-scale uniform Cu-TCPP membranes with an area of 78.5 cm² were successfully prepared by enlarging the filtration area (Supplementary Fig. 33), and the membrane structure remained intact even after being immersed in electrolyte solutions with different pH values for 15 days (Supplementary Fig. 34).

It is important to note that the production efficiency of the vacuum filtration method is quite low and time-consuming. To overcome this limitation, we employed the spraying coating strategy to fabricate a Cu-TCPP membrane with a larger area of 600 cm² at high speed (Detailed conditions described in the method and Supplementary Fig. 36), further highlighting the potential for large-scale production and industrial applications of the 2D Cu-TCPP membrane.

~~The emergence of metal-organic frameworks (MOFs)²¹ and covalent organic frameworks (COFs)²² with intrinsic, regular, and uniformly sized pores has provided new opportunities for enhancing osmotic energy conversion performance. However, the reported MOF membranes for osmotic energy harvesting were mainly constructed by 3D MOF crystals^{23,24}. Achieving optimal osmotic energy conversion efficiency by constructing 2D MOF layer membranes that fully utilize the interlayer nanochannels remains unattainable²⁵. Recent studies have also introduced 2D MOF nanosheets with intrinsic, regular, and uniformly sized pores for constructing MOF layered membranes¹⁷. Leveraging the both the in-plane pores and interlayer transport pathways, these 2D MOF membranes have demonstrated considerable promise in gas and molecular separation¹⁸⁻²¹. However, the utilization of 2D MOF membrane in osmotic energy conversion applications remains notably scarce.~~

In our study, we proposed the construction of 2D nanochannel membranes using MOF nanosheets as building units. Recent studies have introduced effective strategies to enhance the osmotic energy conversion efficiency of traditional 2D lamellar membranes. For instance, integrating 1D materials into 2D lamellar membranes significantly boosts power density while enhancing mechanical strength⁵³. Moreover,

combining materials with different maximum absorption wavelength ranges can further improve light-controlled ion transport efficiency⁵⁴. These innovative approaches provide valuable inspiration for further enhancing the osmotic energy conversion performance of MOF membranes, thereby propelling osmotic energy harvesting from the realm of laboratory research to practical industrial applications.

~~The solutions with salinity gradients are a prerequisite for traditional ionic energy recovery systems, as the salinity gradient is the only driving force for directional ion transport.~~ Although researchers have estimated a substantial 1.0 TW of osmotic energy released globally from the mixing of river water and seawater through theoretical calculations⁵², the practical utilization of osmotic energy presents significant challenges due to the requirement for freshwater or low-concentration solutions, because both of the two prevailing technologies for osmotic energy harvesting, RED technology and Pressure Retarded Osmosis (PRO) technology, require solutions of different concentrations on either side of the membrane.

Furthermore, as the membrane thickness and corresponding testing area increased, we observed a rise in output power attributed to the heightened ion flux. However, due to the substantial accumulation of K^+ ions at the channel exit and the onset of concentration polarization in thicker membranes, a decrease in power density was also noted, consistent with findings from prior studies⁴⁴⁻⁴⁶. Recent reports have indicated this concentration polarization has even been identified as a significant obstacle to the practical implementation of RED technology⁴⁷. It should be noted that when the membrane thickness reached 20 μm and the test area expanded to 0.2 mm^2 , the system achieved a power density of 4.32 W/m^2 (Supplementary Fig. 29 and Table 4). This value not only closely approaches the commercialization benchmark of approximately 5 W/m^2 but also remains competitive when compared to the reported power density achieved with membranes of the same concentration difference and the same area.”

Supplementary Figure 36. Large-scale Cu-TCPP membrane (top: 300 cm², bottom: 600 cm²) fabricated by spraying coating strategy.

Supplementary Figure 29. Output power and power density with different test area. Detailed membrane size parameters are shown in Supplementary Table 4. Error bars indicated the standard deviations from three different samples.

Supplementary Table 4. Thickness and width of membrane for different test areas.

Testing area (10 ⁻² mm ²)	Thickness (μm)	Width (cm)
1	2	0.5
2	4	0.5
4	8	0.5
10	20	0.5
20	20	1.0

Reference:

- (17) Li X, et al. Construction of angstrom-scale ion channels with versatile pore configurations and sizes by metal-organic frameworks. *Nat. Commun.* **14**, 286 (2023).
- (18) Cheng P, et al. Two-dimensional metal-porphyrin framework membranes for efficient molecular sieving. *J. Membrane Sci.* **640**, 119812 (2021).
- (19) Zhou Y, et al. Highly-Efficient Ion Gating through Self-Assembled Two-Dimensional Photothermal Metal-Organic Framework Membrane. *Angew. Chem. Int. Ed.* **62**, e202302997 (2023).
- (20) Li G, et al. Ni-metal-organic-framework (Ni-MOF) membranes from multiply stacked nanosheets (MSNs) for efficient molecular sieve separation in aqueous and organic solvent. *J. Membrane Sci.* **635**, 119470 (2021).
- (21) Song H, et al. Structure Regulation of MOF Nanosheet Membrane for Accurate H₂/CO₂ Separation. *Angew. Chem. Int. Ed.* **62**, e202218472 (2023).
- (44) Pan S, et al. Toward Scalable Nanofluidic Osmotic Power Generation from Hypersaline Water Sources with a Metal–Organic Framework Membrane. *Angewandte Chemie International Edition* **62**, e202218129 (2023).
- (45) Ding L, Xiao D, Zhao Z, Wei Y, Xue J, Wang H. Ultrathin and Ultrastrong Kevlar Aramid Nanofiber Membranes for Highly Stable Osmotic Energy Conversion. *Advanced Science* **9**, 2202869 (2022).
- (46) Gao J, Liu X, Jiang Y, Ding L, Jiang L, Guo W. Understanding the Giant Gap between Single-Pore- and Membrane-Based Nanofluidic Osmotic Power Generators. *Small* **15**, 1804279 (2019).
- (47) Wang L, Wang Z, Patel SK, Lin S, Elimelech M. Nanopore-Based Power Generation from Salinity Gradient: Why It Is Not Viable. *ACS Nano* **15**, 4093–4107 (2021).
- (52) Logan BE, Elimelech M. Membrane-based processes for sustainable power generation using water. *Nature* **488**, 313–319 (2012).
- (53) Zhang Z, Yang S, Zhang P, Zhang J, Chen G, Feng X. Mechanically strong MXene/Kevlar nanofiber composite membranes as high-performance nanofluidic osmotic power generators. *Nature Communications* **10**, 2920 (2019).
- (54) Yeom J, et al. Photosensitive ion channels in layered MXene membranes modified with plasmonic gold nanostars and cellulose nanofibers. *Nature Communications* **14**, 359 (2023).

Review #3

General comment

The authors proposed a bionic nanofluidic membranes with exceptional high ions permeability and power-production ability even under the natural sunlight and symmetric solutions system. Although the membrane performance, in cases of solar/ionic energy conversion, seems impressive, this manuscript could not be published in Nature Communication at the current stage due to the following reasons.

Response to the General Comment

We greatly appreciate the reviewers for their detailed and constructive suggestions. We have tried our best to comply with the suggestions in the revised manuscript. Point-by-point responses to the comments are enclosed.

Comment 1

Regarding the conclusion demonstrated in Intro Part, “horizontal ion transport channels outperformed traditional vertical ion transport channels in 2D nanochannel membranes”, this so-called conclusion is strange for this reviewer. There is no more description on how this conclusion was gained.

Response to Comment 1

We thank the reviewer for the kind comment. We deeply apologize for the confusion caused by our previous statement. Since the concept of constructing nanochannels by stacking 2D nanosheets for precise ion separation was proposed in 2012, 2D nanochannel membranes have ignited substantial research interest. In the realm of RED, there have been numerous reports on optimizing the nanochannel size and surface properties of 2D membranes to enhance their energy conversion efficiency. Recent research has also confirmed that the other geometric characteristics of 2D nanochannel membranes significantly influences ion transport behavior. For instance, regarding the orientation of 2D nanochannels, in most previous studies evaluating 2D layered membranes for osmotic energy conversion, RED performances were assessed using the conventionally used U-shaped device, where ions are transported

perpendicular to the 2D nanochannels, as illustrated in Supplementary Fig. 24. An alternative model has been introduced where ions are transported parallel to the 2D nanochannels. In this configuration, the nanofluidic system exhibited enhanced ion diffusion rates and improved RED performance due to the larger inlet area for ions to enter the internal channel; furthermore, considerably shortened ion transport paths minimized kinetic energy dissipation, further contributing to the system's efficiency (*ACS Nano* 2020, 14, 16654-16662). For instance, Zhang et al. demonstrated that the ion diffusion rate in horizontal 2D GO nanochannels was 13.7 times greater than that in vertical nanochannels (*Adv. Sci.* 2020, 7, 2000286). In the revised manuscript, we have included a clearer description and supplementary schematic figure to explain the differences between the two transport models (Supplementary Fig. 24).

Furthermore, to substantiate the advantages of the horizontal ion transport model over the vertical model, we also conducted experimental investigations to validate that the horizontal model is indeed more effective for ion transport. For instance, under nearly identical salt gradients and a similar test area, even though the membrane length in the horizontal model (3 cm) was significantly longer than the membrane thickness in the vertical model (1 μm), the current in the horizontal model remained nearly consistent with that of the vertical model (Supplementary Figs. 24 and 25). In addition, for the study on the light-responsive ion transport, the horizontal model also provided the possibility to amplify the temperature gradient within the nanochannel, creating a more significant temperature difference to enhance the impact on ion transport efficiency. In the horizontal transportation model, the temperature difference in the Cu-TCPP MOF membrane increased to 120°C in just 90 seconds, whereas in the vertical transportation model, under the same irradiation conditions, the temperature difference was only 5°C (Supplementary Fig. 39). A detailed description has been added in the revised manuscript as follows:

~~“By now, to overcome the trade-off between ion selectivity and permeability, not only the diverse intrinsic physical and chemical properties of various 2D materials was explored¹³, the precise modification of geometry characteristics¹⁴, charge property¹⁵, and wetting behavior¹⁶ of layered nanoarchitecture was also conducted. For example,~~

horizontal ion transport channels have been demonstrated to outperform the traditional vertical direction in 2D nanochannel membranes, effectively shortening the transport path and reducing kinetic energy dissipation of ions, thereby achieving superior RED performance^{17,18}. Until now, many efforts have been made to optimize the properties of nanochannels, including their geometric characteristics and charge properties, to enhance ion selectivity and permeability^{13,14}. Additionally, the significant impact of RED device design on ion transport behavior has also been reported. For example, the vertical model, where ions travel perpendicularly through the 2D nanochannels, has been the predominant choice for evaluating energy conversion efficiency. However, recent advancements in research have shown that when ions are transported parallel to the 2D nanochannels, a higher ion diffusion rate and improved RED performance are achieved due to shortened ion transport paths and minimized kinetic energy dissipation^{15,16}.

Experimental investigations were conducted to validate that the horizontal model is indeed more effective for ion transport than the vertical model. Under nearly identical salt gradients and a similar test area, even though the membrane length in the horizontal model (3 cm) was significantly longer than the membrane thickness in the vertical model (1 μm), the current in the horizontal model nearly matched that of the vertical model, indicating lower resistance in ion transmembrane transport in the horizontal model (Supplementary Figs. 24 and 25).

In addition to the lower transport resistance in the vertical model, the horizontal model also provided the possibility to amplify the temperature gradient within the nanochannel, creating a more significant temperature difference to enhance the impact on ion transport efficiency. To validate this mechanism, we monitored temperature changes in Cu-TCPP membranes under both horizontal and vertical transport modes in the presence of light. In the horizontal model, one end of the membrane was illuminated, and the temperature difference by comparing the illuminated section with the opposite end was measured. In vertical model, we illuminated the front surface of the membrane and measured the temperature difference between the illuminated side and the opposite side of the membrane (Supplementary Fig. 39). The temperature

difference in the Cu-TCPP membrane increased to 120°C in just 90 seconds in the horizontal transportation model, whereas in the vertical transportation model, under the same irradiation conditions, the temperature difference was only 5°C.”

Supplementary Figure 24. Schematic illustration two different transport models (a) and the membrane size for ionic current comparison (b).

Supplementary Figure 25. Stabilized ionic currents in the horizontal and vertical transport model.

Supplementary Figure 39. Schematic of the temperature measurement for two models for under illumination. (a) In the horizontal model, the temperatures of illuminated and unilluminated ends were measured by infrared thermographic camera. (b) In the vertical model, the temperature of illuminated surface was measured by infrared thermographic camera, and the opposite side of the membrane was monitored by connecting to a thermocouple.

Reference:

- (13) Zhang Z, Wen L, Jiang L. Bioinspired smart asymmetric nanochannel membranes. *Chem. Soc. Rev.* 47, 322-356 (2018).
- (14) Li S, Wang J, Lv Y, Cui Z, Wang L. Nanomaterials-Based Nanochannel Membrane for Osmotic Energy Harvesting. *Adv. Funct. Mater.* 2308176.
- (15) Wen Q, et al. Electric-Field-Induced Ionic Sieving at Planar Graphene Oxide Heterojunctions for Miniaturized Water Desalination. *Adv. Mater.* 32, 1903954 (2020).
- (16) Qu R, et al. Vertically-Oriented $Ti_3C_2T_x$ MXene Membranes for High Performance of Electrokinetic Energy Conversion. *ACS Nano* 14, 16654-16662 (2020).

Comment 2

Other researches also reported the MOF membranes for energy conversion. The authors only assigned the difference to the enhanced completely utilized interlayer nanochannels for ions transport. This reviewer thinks with conservative gratitude that this alone is not sufficient. From the novelty of perspective, the authors are recommended to initiate some more points to strengthen the superiority and uniqueness of this research.

Besides, this reviewer is concerned the true differentiations and superiorities of interlayer transport channels when compared to that of MOF membranes which may have its own intra-/inter-crystalline defects for ions high-speed transportation.

Response to Comment 2

We thank the reviewer for the kind comment. Point-by-point responses to the comments are enclosed as follows:

1. For the difference with reported MOF membrane for energy conversion

Based on the literature review, we found that the MOF membranes reported for osmotic energy recovery predominantly possess a 3D structure (*Nano Energy*, 2018, 53, 643-649; *Sci. Adv.*, 2021, 7, eabe9924; *ACS Nano*, 2022, 16, 16343-16352). However, the application of 2D MOF membranes in osmotic energy recovery is notably absent. Therefore, in our study, we opted for an approach by employing 2D MOF stacking to create a layered membrane for osmotic energy recovery. In addition, despite the successful preparation of several single-layer 2D MOFs such as Zn₂(bim)₄, Zr-BTB and Cu-BDC as reported by researchers (*Science*, 2014, 346(6215, 1356-1359; *J. Am. Chem. Soc.*, 2016, 138, 6636-6642; *Chem. Commun.*, 2017, 53, 4254-4257), we specifically chose Cu-TCPP MOF due to its exceptional photothermal response characteristics because we consider that the introduction of light to induce a temperature field within the nanochannels will provide a robust driving force for ion transport to efficiently enhance energy conversion efficiency.

Besides taking advantage of the unique porous structure and photothermal conversion performance, the choice of this 2D membrane model is further supported by the following rationale:

1. Chemical modifications of nanochannel can be conducted in the solution phase before preparing the 2D membrane, markedly enhancing efficiency of channel geometry and chemical properties modification (*Angew. Chem. Int. Ed.*, 2023, 62, e202218321; *ACS Nano*, 2019, 13, 4238-4245);
2. When combined with 0D or 1D nanomaterials to construct heterogeneous structures, the synergistic exploitation of different materials' inherent advantages can further enhance osmotic energy conversion performance (*Nat. Commun.*, 2023, 14, 359; *J. Am. Chem. Soc.*, 2021, 143, 16206-16216);
3. It is facile to create more sophisticated biomimetic structures to achieve a novel

ion transport behavior (*Angew. Chem. Int. Ed.* 2022, 61, e202206152).

The objective of this study was to initially demonstrate the feasibility of efficient osmotic energy retrieval using 2D MOF membranes. In our future research, we will try to unlock the complete potential of 2D MOF materials via the strategies as described above to further enhance energy conversion efficiency further.

2. For the effect of intra-/inter-crystalline defects on ion transport

As noted by the reviewer, two types of structure defect may exist in 3D MOF membrane: intra- and inter-crystalline defects (*Acc. Mater. Res.*, 2022, 3, 735-747). Specifically, poor inter-growth quality during MOF membrane preparation results in inter-crystalline defects. In the synthesizing MOF nanosheet process, missing linkers or missing metal clusters result in intra-crystalline defects. These defects enhance membrane flux but simultaneously reduce its selectivity. Recent reports suggest that structural defects of 3D MOF membrane could be regulated by controlling the preparation conditions, thereby optimizing the selectivity and permeability performance (*J. Phys. Chem. C*, 2019, 123, 16118-16126; *Nat. Commun.*, 2022, 13, 266).

For the Cu-TCPP membrane, although the similar structural defects are inevitable, we consider that the consistency in the conditions for MOF nanosheet preparation and membrane fabrication in our study ensures that the influence of these defects on ion transport behavior is uniform in different experiment batches. Furthermore, it may be due to current limitations in characterization methods, accurately characterizing defects in 2D MOF nanosheets presents challenges. As a result, there is limited literature addressing intra- and inter-crystalline defects in 2D MOF membranes for high-speed ion transport. However, drawing inspiration from the insightful suggestions provided by the reviewer, we have identified that defects within MOF nanosheets could potentially serve as a unique means to modulate ion transport behavior and enhance osmotic energy conversion performance. We are enthusiastic about exploring deeper into this area, conducting thorough research to unlock the full potential of 2D MOF membranes in energy harvesting. A detailed description has been added in the revised manuscript as follows:

~~“The emergence of metal-organic frameworks (MOFs)²¹ and covalent organic frameworks (COFs)²² with intrinsic, regular, and uniformly sized pores has provided new opportunities for enhancing osmotic energy conversion performance. However, the reported MOF membranes for osmotic energy harvesting were mainly constructed by 3D MOF crystals^{23,24}. Achieving optimal osmotic energy conversion efficiency by constructing 2D MOF layer membranes that fully utilize the interlayer nanochannels remains unattainable²⁵. Recent studies have also introduced 2D MOF nanosheets with intrinsic, regular, and uniformly sized pores for constructing MOF layered membranes¹⁷. Leveraging the both the in-plane pores and interlayer transport pathways, these 2D MOF membranes have demonstrated considerable promise in gas and molecular separation¹⁸⁻²¹. However, the utilization of 2D MOF membrane in osmotic energy conversion applications remains notably scarce.~~

In our study, we proposed the construction of 2D nanochannel membranes using MOF nanosheets as building units. Recent studies have introduced effective strategies to enhance the osmotic energy conversion efficiency of traditional 2D lamellar membranes. For instance, integrating 1D materials into 2D lamellar membranes significantly boosts power density while enhancing mechanical strength⁵³. Moreover, combining materials with different maximum absorption wavelength ranges can further improve light-controlled ion transport efficiency⁵⁴. These innovative approaches provide valuable inspiration for further enhancing the osmotic energy conversion performance of MOF membranes, thereby propelling osmotic energy harvesting from the realm of laboratory research to practical industrial applications.”

Reference:

- (17) Li X, et al. Construction of angstrom-scale ion channels with versatile pore configurations and sizes by metal-organic frameworks. *Nat. Commun.* **14**, 286 (2023).
- (18) Cheng P, et al. Two-dimensional metal-porphyrin framework membranes for efficient molecular sieving. *J. Membrane Sci.* **640**, 119812 (2021).
- (19) Zhou Y, et al. Highly-Efficient Ion Gating through Self-Assembled Two-Dimensional Photothermal Metal-Organic Framework Membrane. *Angew. Chem. Int. Ed.* **62**, e202302997 (2023).
- (20) Li G, et al. Ni-metal-organic-framework (Ni-MOF) membranes from multiply stacked nanosheets (MSNs) for efficient molecular sieve separation in aqueous and organic solvent. *J. Membrane Sci.* **635**, 119470 (2021).
- (21) Song H, et al. Structure Regulation of MOF Nanosheet Membrane for Accurate H₂/CO₂ Separation.

Angew. Chem. Int. Ed. **62**, e202218472 (2023).

(53)Zhang Z, Yang S, Zhang P, Zhang J, Chen G, Feng X. Mechanically strong MXene/Kevlar nanofiber composite membranes as high-performance nanofluidic osmotic power generators. *Nat. Commun.* **10**, 2920 (2019).

(54)Yeom J, et al. Photosensitive ion channels in layered MXene membranes modified with plasmonic gold nanostars and cellulose nanofibers. *Nat. Commun.* **14**, 359 (2023).

Comment 3

From the AFM and XRD results, how could the conclusion of “the ions exhibited a tendency to diffuse within the interlayer spacing between the few-layer Cu-TCPP nanosheets rather than between the monolayer Cu-TCPP”. The relevant interpretation is rather ambiguous and confusing.

Response to Comment 3

We thank the reviewer for the insightful comment. Based on a literature review, we found that the thicknesses of most reported TCPP-based MOF nanosheets, as measured by AFM, fall within the range of 2-5 nm, as documented in the following **Table R1**. A limited number of studies have reported that the thickness of the prepared nanosheets closely approximates the theoretical thickness of 0.27 nm. In our present study, the thickness of the Cu-TCPP nanosheets prepared was approximately 2.3 nm, as determined through AFM measurements.

Table R1. Thickness of reported 2D porphyrin-based MOF nanosheets.

Material	Thickness of nanosheets (nm)	Refs.
PMOF	1.9	1
Al-MOF	1.9	2
Cu-TCPP	2.0	3
Cu-TCPP	4.0	4
Al-TCPP	17±1.2	5
Cu-TCPP	0.8	6
Zn-TCPP	7.6±2.6	
Cu-TCPP	4.5±1.2	7
Co-TCPP	18.8±6.4	
Zn-TCPP(Fe)	7.4±2.9	
Cu-TCPP	5.4	8
MOF	4~5	9
Cu-TCPP	3.0	10
Cu-TCPP	4~5	11

Cu-TCPP	1.5±0.5	12
Co-TCPP	40	13
Cu-TCPP(Fe)	1.0	14
Zn-TCP(Fe)	1.9±0.1	15
Cu-TCPP	25	16
In-TCPP	3.97	17
M-TCPP	7.7±3.5	18
Cu-TCPP	6.5	
Co-TCPP	4.8	19
Ce-TCPP	3.7	
Cu-TCPP	2.9	20
Pd-TCPP	1.38±0.24	21
MPFs	2.96	22
Cu-TCPP	2.6	23
Cu-TCPP	10	24
Cu-TCPP	5.25	25
Cu-TCPP(Fe)	10	26
Zn-TCPP	2.7	27
Cu-TCPP	20	28
Cu-TCPP	2.5	29
Zn-TCPP	Less than 5	30
Cu-TCPP	0.27	31
Cu-TCPP	2.3	This work

Reference:

(1) Zhang Y, *et al. CCS Chem.* **4**, 3329-3341 (2022); (2) Jian M, *et al. Sci. Adv.* **6**, eaay3998 (2020); (3) Hussain S, *et al. J. Membrane Sci.* **620**, 118888 (2021); (4) Chen J, *et al. J. Membrane Sci.* **648**, 120382 (2022); (5) Xiao Y, *et al. Micropor. Mesopor. Mat.* **324**, 111272 (2021); (6) Cai J, *et al. Nano Res.* **16**, 6290-6297 (2023); (7) Zhao M, *et al. Adv. Mater.* **27**, 7372-7378 (2015); (8) Zhou Y, *et al. Angew. Chem. Int. Ed.* **62**, e202302997 (2023); (9) Zhao X, *et al. Nano Energy* **81**, 105657 (2021); (10) Cheng P, *et al. J. Membrane Sci.* **640**, 119812 (2021); (11) Wu G-D, *et al. Angew. Chem. Int. Ed.* **60**, 9931-9935 (2021); (12) Hussain S, *et al. Sep. Purif. Technol.* **268**, 118688 (2021); (13) Chen L, *et al. Nanoscale* **11**, 11121-11127 (2019); (14) Wang C, *et al. Adv. Funct. Mater.* **30**, 1908804 (2020); (15) Ang H, *et al. ACS Appl. Mater. Interfaces* **9**, 28079-28088 (2017); (16) Zhao S, *et al. Environ. Sci. Pollut. R.* **27**, 39186-39197 (2020); (17) Zhang Z, *et al. Nanoscale* **14**, 7146-7150 (2022); (18) Ding G, *et al. Adv. Funct. Mater.* **29**, 1806637 (2019); (19) Li S, *et al. Mol. Catal.* **496**, 111196 (2020); (20) Liang Y, *et al. Sep. Purif. Technol.* **282**, 120045 (2022); (21) Zhang H, *et al. Chem. Eng. J.* **442**, 136144 (2022); (22) Zhao W, *et al. Small* **15**, 1901351 (2019); (23) Wang Z, *et al. J. Membrane Sci.* **633**, 119397 (2021); (24) Wang C, *et al. Chem.* **835**, 178-185 (2019); (25) Li B, *et al. Theranostics* **8**, 4086-4096 (2018); (26) Chen H, *et al. ACS Appl. Mater. Interfaces* **10**, 24108-24115 (2018); (27) Zhang L, *et al. Angew. Chem. Int. Ed.* **60**, 22664-22671 (2021); (28) Xu G, *et al. J. Am. Chem. Soc.* **134**, 16524-16527 (2012); (29) Jian M, *et al. Chem. Commun.* **53**, 13161-13164 (2017); (30) Li X, *et al. ACS Appl. Mater. Interfaces* **11**, 24618-24626 (2019); (31) Yang D, *et al. Chem. Int. Ed.* **59**, 18954-18959 (2020).

The precision of AFM measurements could be influenced by adsorbed surfactants on both flakes and substrates (*Nat. Commun.*, 2014, 5, 4576). For example, in our prior research, the height measured for $\text{Ti}_3\text{C}_2\text{T}_x$ nanosheets, with a theoretical thickness of 0.88 nm, was around 1~1.2 nm under our experimental conditions (*Energy Environ. Sci.*, 2023, 16, 3146-3157). Nevertheless, even after accounting for potential errors stemming from surface adsorption, it is evident that the thickness of the Cu-TCPP nanosheets we prepared significantly exceeds their theoretical thickness (*Angew. Chem. Int. Ed.* 2020, 59, 18954). Consequently, we consider the nanosheets produced in this study to be characterized as few-layered nanosheets rather than truly single-layer nanosheets.

To address the reviewer's concern, we have conducted the Wide-Angle X-ray Scattering (WAXS) measurement for a comprehensive structural evaluation of the 2D Cu-TCPP membrane, and the out-of-plane WAXS pattern of the MOF membrane was selected to confirm the oriented packing of the nanosheets (Supplementary Figs. 5 and 6). The peak at 19° , which could be also found in XRD pattern of bulk Cu-TCPP crystal reported previously (*CCS Chemistry*, 2022, 4, 3329-3341; *J. Am. Chem. Soc.*, 2013, 135, 7438-7441), corresponded to a d-spacing of 4.7 Å between the single-layer nanosheets within the multi-layered nanosheets. Furthermore, the presence of two peaks at 2θ at 9° and 12° , absent in the Cu-TCPP crystal, indicated the d-spacing of the stacked multi-layer nanosheets as 0.97 nm and 0.73 nm. Similar results were found in previously reported 2D MOF membranes. For example, in Yang et al.'s study, the XRD pattern of the 2D $\text{Zn}_2(\text{bim})_4$ MOF membrane displayed a peak representing the ordered pristine bulk $\text{Zn}_2(\text{bim})_4$ crystal structure, along with an additional peak at low angles due to the expanded restacking of nanosheets (*Science*, 2014, 346, 1356-1359).

As a result, the size between multi-layer nanosheets, calculated by subtracting the theoretical thickness of one layer of Cu-TCPP, is approximately 0.17 nm. It presents a significant challenge for ions to achieve transmembrane diffusion within this confined space. However, the channels between the multi-layer nanosheets, with a channel size ranging from 0.6 to 0.7 nm, offer the necessary space for rapid ions transport. A

detailed description has been added in the revised manuscript as follows:

~~“According to previous studies, the theoretical thickness of monolayer Cu-TCPP nanosheet is 0.28 nm³⁶, while the average thickness of the prepared Cu-TCPP nanosheet, as determined via atomic force microscopy (AFM), were approximately 2.3 nm (Fig. 2d and Supplementary Fig. 4), indicating that the Cu-TCPP nanosheet prepared in our study consist of few layers of monolayer nanosheets. To further analyze the structural characteristics of the obtained few-layer Cu-TCPP nanosheet, we conducted the X-ray diffraction (XRD) tests on the in-plane and out-of-plane structures. The peak in the out-of-plane XRD pattern at ~19.6° indicated a d-spacing of 4.5 Å for the two adjacent Cu-TCPP monolayers, consistent with previous literature (Supplementary Figs. 5 and 6)^{27,37}~~

~~Combining with the AFM and XRD results presented above, we infer that ions exhibit a tendency to diffuse within the interlayer spacing between the few-layer Cu-TCPP nanosheets rather than between the monolayers Cu-TCPP. Additionally, the large number of regular nanopores in the MOF nanosheets could provide "shortcuts" for ion transport, which could enhance the ion flux^{40,41}~~

Previous studies have indicated that the theoretical thickness of a monolayer Cu-TCPP nanosheet is 0.27 nm³³. However, the average thickness of the prepared Cu-TCPP nanosheet, as determined via atomic force microscopy (AFM), measured approximately 2.3 nm (Fig. 2d and Supplementary Fig. 4). The precision of AFM measurements could be influenced by adsorbed surfactants on both flakes and substrates³⁴. However, even after accounting for errors caused by surface adsorption, the thickness of the Cu-TCPP nanosheets we prepared far exceeded their theoretical thickness. Hence, we conclude that the nanosheets prepared in this study are multi-layered rather than truly single-layered nanosheets. The Wide-Angle X-ray Scattering (WAXS) analysis was conducted to further elucidate the structural characteristics of the obtained multi-layer Cu-TCPP nanosheet (Supplementary Figs. 5 and 6). In the out-of-plane WAXS pattern of the MOF membrane, the peak at 19°, which also appeared in the XRD pattern of bulk Cu-TCPP crystal reported previously^{23,35}, corresponded to a d-spacing of 4.7 Å between the single-layer

nanosheets within the multi-layered nanosheets. Moreover, the presence of two distinct peaks at $2\theta = 9^\circ$ and 12° , absent in the Cu-TCPP crystal, indicated the d-spacing of the stacked multi-layer nanosheets as 0.97 nm and 0.73 nm, respectively. Similar findings have been reported in previous studies on 2D MOF membranes. For instance, in Yang et al.'s research, the XRD pattern of the 2D $\text{Zn}_2(\text{bim})_4$ MOF membrane exhibited peaks representing the ordered pristine bulk $\text{Zn}_2(\text{bim})_4$ crystal structure, along with peaks at low angles due to the expanded restacking of nanosheets³⁶. Based on these observations, the size between multi-layer nanosheets, calculated by subtracting the theoretical thickness of one layer of Cu-TCPP, have an approximate diameter of 0.17 nm, presenting a significant challenge for ions to achieve transmembrane diffusion within this confined space. However, the nanochannels located between the multi-layer nanosheets, with channel size of approximately 0.6 nm, offer the necessary space for rapid ions transport.”

Supplementary Figure 5. Out-of-plane WAXS pattern of Cu-TCPP lamellar membrane.

Reference:

- (23) Zhang Y, et al. Bidirectional Light-Driven Ion Transport through Porphyrin Metal–Organic Framework-Based van der Waals Heterostructures via pH-Induced Band Alignment Inversion. *CCS Chem.* 4, 3329-3341 (2022).
- (33) Yang D, Zuo S, Yang H, Zhou Y, Wang X. Freestanding millimeter-scale porphyrin-based monoatomic layers with 0.28 nm thickness for CO_2 electrocatalysis. *Angew. Chem. Int. Ed.* 59, 18954-18959 (2020).
- (34) Backes C, et al. Edge and confinement effects allow in situ measurement of size and thickness of liquid-exfoliated nanosheets. *Nat. Commun.* 5, 4576 (2014).
- (35) Xu G, Otsubo K, Yamada T, Sakaida S, Kitagawa H. Superprotonic conductivity in a highly oriented crystalline metal–organic framework nanofilm. *J. Am. Chem. Soc.* 135, 7438-7441 (2013).

Comment 4

This reviewer is confused about the statements of high ions selectivity of the as-performed bionic membranes. The high ions selectivity performance was not eye-catching in this manuscript. Therefore, the reader could not figure out the target ions pairs which might be separated by the stimulus-response Cu-TCPP nanosheet membrane. The underlying mechanism of the ions separation mechanisms should be strengthened. According to the limited knowledge of this reviewer and the presented information in this manuscript, the interlayer space of the as formed Cu-TCPP was 0.45 nm, and the intrinsic pore of the Cu-TCPP membrane was 0.84 nm. Both of the slit pore and the in-plane pore were incapable of separating ions. In addition, there are much more 2D MOF COF HOF materials, why the authors chose the Cu-TCPP materials as the main research target, how about other porous materials when compared to that of Cu-TCPP.

Response to Comment 4

We thank the reviewer for the kind comment. Point-by-point responses to the comments are enclosed as follows:

1. For the ion selectivity of Cu-TCPP membrane

In our study, we initially confirmed the ion selectivity of the 2D Cu-TCPP membrane through the observed nonlinear relationship between ionic conductance and corresponding concentration. Specifically, zeta potential measurements of the nanosheets (Figure 2e and Supplementary Fig. 8) revealed the negatively charged nature of the Cu-TCPP nanosheet surfaces. According to classical Electrical Double Layer (EDL) theory, an EDL region forms at the interface between the charged channel wall and the ionic solution due to electrostatic attraction to counterions and repulsion from co-ions. The Debye length (λ_D), characterizing the decay of electrostatic forces as a result of screening the surface charges with counter ions in electrolyte solution, typically ranges from tens of nanometers down to Angstroms as per the Poisson-Boltzmann theory (*Chem. Soc. Rev.*, 2010, 39, 1073–1095). In micrometer-sized channels, the EDL thickness is negligible compared to channel dimensions, making the effect of surface charge potential on ion transport behavior

less evident. However, based on our WAXS results described above, the interlayer channel diameter in the Cu-TCPP membrane was confined to the nanometric scale. In this scenario, the spaces occupied by EDLs in the extremely confined Cu-TCPP channel could not be neglected. The EDLs at opposite surfaces might even overlap in the channels due to the nanochannel diameter being less than twice the Debye length, suggesting that ion population and distribution in the nanochannel were strongly influenced by the channel surface charge (*Nat. Rev. Mater.*, 2021, 6, 622-639; *Nat. Nanotech.*, 2022, 17, 622-628; *J. Am. Chem. Soc.*, 2012, 134, 16528-16531). As a result, the Cu-TCPP membrane exhibited outstanding separation ability towards cations and anions.

Additionally, we conducted immersion experiments to further investigate the membrane's ion selectivity as described in previous study (*Adv. Funct. Mater.*, 2017, 27, 1603623). The Cu-TCPP membrane was immersed in a 0.5 M KCl solution, after which its surface was subjected to EDS scanning (Supplementary Fig. 20). The higher adsorption amount of K^+ compared to Cl^- on the MOF membrane's surface further confirmed that the negatively charged MOF nanochannels selectively attracted counterions due to electrostatic forces. Furthermore, the cation transfer number (t_+) of the Cu-TCPP membrane was calculated, as presented in Supplementary Fig. 22 and Table 1. The consistently greater than 0.85 values of t_+ under different concentration gradients indicated the overall cation-selective characteristics of the Cu-TCPP membrane.

2. For the ion diffusion through intrinsic pores and interlayer channels

Regarding the interlayer channels: As previously indicated in **Response to Comment 3**, the 0.45 nm distance represents the d-spacing between single-layer nanosheets within the multi-layered nanosheets prepared in our study. It is challenging for ions to pass through this interlayer channel. The WAXS analysis also confirmed the existence of a nanochannel between multi-layer nanosheets, and the ions tend to diffuse across the membrane through this space.

Regarding the in-plane channels: The intrinsic pore size of the Cu-TCPP nanosheet was found to be 0.84 nm, slightly larger than the hydration diameter of ions such as

Na⁺ and K⁺ that were investigated in our study. In an aqueous environment, a hydration shell formed on the ion's surface due to electrostatic effects. Recent studies have revealed that ions within nanochannels cannot be simply regarded as rigid spheres. The surrounding hydration shell could deform or even be removed, allowing ions to permeate through nanochannels with dimensions comparable to or even smaller than the ion diameter (*Nat. Nanotech.*, 2017, 12, 546-550).

3. For the advantages of MOF towards COF and HOF

During the experimental design, we adhered to two crucial criteria for selecting the MOF material to construct the nanochannel membrane. The chosen material needed to be capable of facile and cost-effective preparation in a 2D structure and demonstrate excellent photothermal properties. For Cu-TCPP MOF, the ultrathin 2D nanosheets could be easily prepared by controlling crystal growth with the assistance of surfactants (*Adv. Mater.* 2015, 27, 7372-7378), and its photothermal conversion property had been previously reported in cancer phototherapy (*J. Inorg. Biochem.*, 2021, 225, 111599; *Theranostics* 2018, 8, 4086). Therefore, Cu-TCPP was chosen as the building block for the preparation of the light-responsive nanochannel membrane. Covalent organic frameworks (COFs) represent another category of crystalline porous materials formed by the covalent bonding of organic molecules. The preparation conditions of COFs reported for osmotic energy harvesting were summarized in the follow **Table R2**. Like 2D MOFs, 2D COF nanosheets are popular candidates for membrane preparation. However, the preparation of COFs was relatively complicated and time-consuming as shown in **Table R2**.

Table R2. Comparison of preparation time of reported COF membranes.

Monomer A	Monomer B	Preparation time	Ref.
1,3,5-triformylphloroglucinol (TFP)	4-aminophenyl)amine (TPA-3NH ₂)	48h	1
1, 3, 5-Triformylphloroglucinol (Tp)	2, 5-diaminobenzenesulfonic acid (PaSO ₃ H)	72h	2
1,3,5-tris(4-aminophenyl)-benzene (TAPB)	2,5-dihydroxyterephthalic acid (DHTA)	24h	3

2,5-bis(2-(dimethylamino)ethoxy)terephthalohydrazide (BTH)	2,2'-((2,5-di(hydrazinecarbonyl)-1,4-phenylene)bis(oxy))bis(N,N,N-trimethylethan-1-aminium) iodide (BTA)	3d	4
tetraphenylmethaneamine (TAM)	terephthalaldehyde (TPA)	3d	5
1,3,5-triformylphloroglucinol (Tp)	2,5-diaminobenzenesulfonic acid (Pa-SO ₃ H)	6d	6
Triformylphloroglucino (Tp)	Sodium 2,5-diaminobenzenesulfonate	3d	7
tetrakis(p-formylphenyl)pyrene (TFPPY)	2,5-diaminebenzenesulfonic acid (Pa-SO ₃ H)	7d	8
triformylphloroglucinol (Tp)	ethidium bromide (EB)	4d	9
5,10,15,20-Tetrakis(4-aminophenyl)porphyrin (TAPP)	2,5-dihydroxyterephthalaldehyde	5d	10
2,4,6-trimethyl-1,3,5-triazine (TMT)	1,3,5-tris(4-formylphenyl)triazine (TFPT)	3d	11
2,4,6-trimethyl-1,3,5-triazine (TMT)	1,3,5-triformylbenzene (TB)	3d	
2,5-bis(2-(dimethylamino)ethoxy)terephthalohydrazide (BTH)	triformylphloroglucinol (Tp)	3d	12
tetra-(4-carboxyphenyl) porphyrin	Copper nitrate trihydrate	6h	This work

Reference:

(1) Gao M, *et al. Nano Energy*, **105**, 108007 (2023); (2) Cao L, *et al. ACS Nano*, **16**, 18910-18920 (2022); (3) Cao L, *et al. J. Am. Chem. Soc.*, **144**, 12400-12409 (2022); (4) Zhu C, *et al. ACS Energy Lett.*, **7**, 2937-2943 (2022); (5) Chen S, *et al. J. Am. Chem. Soc.*, **143**, 9415-9422 (2021); (6) Hou S, *et al. Angew. Chem. Int. Ed.*, **133**, 10013-10018 (2021); (7) Zuo X, *et al. Angew. Chem. Int. Ed.*, **61**, e202116910 (2022); (8) Man Z, *et al. J. Am. Chem. Soc.*, **143**, 16206-16216 (2021); (9) Xian W, *et al. Nat. Commun.*, **13**, 3386 (2022); (10) Zhang Z, *et al. Nat. Commun.*, **13**, 3935 (2022); (11) Wang K, *et al. J. Am. Chem. Soc.*, **145**, 5203-5210 (2023); (12) Zhu C, *et al. Adv. Funct. Mater.* **32**, 2109210 (2022);

Hydrogen-bonded organic frameworks (HOFs) constitute a novel type of porous materials assembled through hydrogen bonding interactions among organic units. They are considered to have significant potential in the preparation of nanochannel membranes due to their advantages of high surface area, simple synthesis, predictable structure, and tailored pore size. Jiang et al. first proposed using HOFs as structural units to create membranes and explored their proton transport characteristics and

application potential in fuel cells (*J. Membrane Sci.*, 2022, 664, 121118). However, current research on HOF membranes is still in its initial stage. This is due to the weak hydrogen bonding in HOFs, which makes it challenging to stabilize the structure directly, thereby hindering the formation of a stable membrane structure. A detailed description has been added in the revised manuscript as follows:

“According to classical Electrical Double Layer (EDL) theory, an EDL region forms at the interface between the charged channel wall and the ionic solution due to electrostatic attraction to counterions and repulsion from co-ions. The Debye length (λ_D), characterizing the decay of electrostatic forces as a result of screening the surface charges with counter ions in electrolyte solution, typically ranges from tens of nanometers down to Angstroms as per the Poisson-Boltzmann theory⁴⁰. In micrometer-sized channels, the EDL thickness is negligible compared to channel dimensions, making the effect of surface charge potential on ion transport behavior less evident. However, based on our XRD and TEM results described above, the interlayer channel diameter in the Cu-TCPP membrane was confined to the nanometric scale. In this scenario, the spaces occupied by EDLs in the extremely confined Cu-TCPP channel could not be neglected, especially at low ionic concentrations. As a result, ion population and distribution in the nanochannel were strongly influenced by the channel surface charge, and the Cu-TCPP membrane exhibited outstanding separation ability towards cations and anions. Similar behavior had been observed in other 2D nanofluidic systems, suggesting the excellent cation/anion separation ability of Cu-TCPP membrane^{2,9,39,41}. Additionally, we conducted immersion experiments to further illustrate the membrane's ion selectivity. The Cu-TCPP membrane was immersed in a 0.5 M KCl solution, after which its surface was subjected to EDS scanning (Supplementary Fig. 20). The higher adsorption amount of K^+ compared to Cl^- on the Cu-TCPP membrane's surface further confirmed that the negatively charged Cu-TCPP nanochannels selectively attracted counterions due to electrostatic forces.”

Supplementray Figure 20. EDS mappings images of K^+ and Cl^- and content information of membrane surface after immersing in 0.5 M KCl solution for 10 h. Scale bar, 10 μ m.

Reference:

- (2) Zhang Z, Wen L, Jiang L. Nanofluidics for osmotic energy conversion. *Nat. Rev. Mater.* 6, 622-639 (2021).
- (9) Yang J, et al. Advancing osmotic power generation by covalent organic framework monolayer. *Nat. Nanotech.* 17, 622-628 (2022).
- (39) Raidongia K, Huang J. Nanofluidic ion transport through reconstructed layered materials. *J. Am. Chem. Soc.* 134, 16528-16531 (2012).
- (30) Bocquet L, Charlaix E. Nanofluidics, from bulk to interfaces. *Chem. Soc. Rev.* 39, 1073-1095 (2010).
- (41) Qin S, et al. High and stable ionic conductivity in 2D nanofluidic ion channels between boron nitride layers. *J. Am. Chem. Soc.* 139, 6314-6320 (2017).

Comment 5

The true seawater and river water other than the model seawater are recommended. Truly, the NaCl was not the sole component in the real circumstance. Besides, the economic-technologic analysis are strongly recommend to be added in this manuscript.

Response to Comment 5

We appreciate the reviewer's kind suggestion. We conducted experiments using real seawater to evaluate the osmotic energy conversion performance of the Cu-TCPP membrane under practical conditions. The main composition and concentrations of

seawater from the Bohai Sea (China) is presented in Supplementary Table 5. Our experimental results demonstrated that the power density reached 13.04 W/m² under actual seawater conditions (Supplementary Figure 32). Recent studies have revealed the reduced membrane permeability in practical long-term energy harvesting processes. These effects are attributed to membrane fouling phenomena resulting from the presence of pollutants in real seawater (*ACS Nano* 2021, 15, 5838-5860; *Angew. Chem. Int. Ed.* 2021, 60, 1-8). Therefore, to advance osmotic energy harvesting technology from the experimental phase to practical applications, elucidating the mechanisms of membrane fouling and developing anti-fouling strategies will be an important topic in our future research.

In addition, as suggested by the reviewer, we calculated the cost of the Cu-TCPP membrane and compared it with other commercial membranes utilized in reverse electro dialysis (*J. Membrane Sci.* 2009, 343, 7-15; *J. Power Sources* 2021, 511, 230460; *J. Membrane Sci.* 2020, 595, 117585; *Membranes* 2020, 10, 168) for an economic-technologic analysis. Considering the widely used thickness of 4 μm in our experiments, the cost per square meter of Cu-TCPP membrane is approximately 300 RMB, which is at least 5 times lower than the price of commercial membranes. This cost comparison confirms the significant economic advantages of the Cu-TCPP membrane. A detailed description has been added in the revised manuscript as follows:

“Real seawater from Bohai Sea (China) was also used to evaluate the osmotic energy conversion performance of the Cu-TCPP membrane under practical conditions (Supplementary Table 5). The power density of Cu-TCPP membrane reached 13.04 W/m² under actual seawater conditions (Supplementary Fig 32).

A cost comparison between the Cu-TCPP membrane and other commercial membranes utilized in RED was also conducted (Supplementary Table 6). Considering the widely used Cu-TCPP membrane with a thickness of 4 μm in our experiments, the cost per square meter is approximately 300 RMB, which is at least 5 times lower than the price of commercial membranes.”

Supplementary Table 5. Main composition and concentrations of seawater.

Type	Na (mg/L)	K (mg/L)	Mg (mg/L)	Ca (mg/L)
Bohai Sea	9350	334	1056	371

**Supplementary Figure 32. Osmotic energy conversion performance of Cu-TCPP membrane using seawater from the Bohai Sea (China).****Supplementary Table 6. Price List of materials elected commercial membranes.**

Material	Company	Price
Cu(NO ₃) ₂ ·3H ₂ O	Macklin, China	0.11 CNY/g
Tetra-(4-carboxyphenyl) porphyrin	Macklin, China	76 CNY/g
N,N-Dimethylformamide	Aladdin, China	45.36 CNY/L
Ethanol	Greagent, China	14.31 CNY/L

Membrane	Company	Price
Fumasep FKD	Fumatech, Germany	16633 CNY/m ²
Selemion CMV	Asahi Glass Company, Japan	30000 CNY/m ²
Fuji CEM-II	Fujifilm, Japan	1487.6 CNY/m ²
Nafion 117	Dupont, America	22950.8 CNY/m ²

The data of Cu(NO₃)₂·3H₂O and tetra-(4-carboxyphenyl) porphyrin were collected from www.macklin.cn, visited in Oct. 23th, 2023. The data of DMF were collected from www.aladdin-e.com, visited in Oct. 23th, 2023. The data of ethanol were collected from titansci.com, visited in Oct. 23th, 2023. The prices of commercial cation exchange membranes were collected from www.1688.com, visited in Oct. 23th, 2023.

Comment 6

It would be better for the authors to show the relationship between the optimized membrane thickness and the best ions transport/energy conversion properties.

Response to Comment 6

We thank the reviewer for the kind comment. As recommended by the reviewer, we conducted an investigation into the influence of Cu-TCPP membrane thickness on osmotic energy power density. It should be noted that recent reports have indicated that applying large-area nanochannel membranes to practical osmotic energy harvesting only leads to a marginal increase in output power (*Angew. Chem. Int. Ed.* 2023, 62, e202218129). Our results also revealed a similar decrease in power density as membrane thickness gradually increased, which could be due to the significant accumulation of K⁺ ions at the channel exit and the occurrence of concentration polarization due to the increased ion flux (*Small* 2019, 1804279. *ACS Nano* 2021 4093-4107). A detailed description has been added in the revised manuscript as follows:

“Furthermore, as the membrane thickness and corresponding testing area increased, we observed a rise in output power attributed to the heightened ion flux. However, due to the substantial accumulation of K⁺ ions at the channel exit and the onset of concentration polarization in thicker membranes, a decrease in power density was also noted, consistent with findings from prior studies⁴⁴⁻⁴⁶. Recent reports have indicated this concentration polarization has even been identified as a significant obstacle to the practical implementation of RED technology⁴⁷. It should be noted that when the membrane thickness reached 20 μm and the test area expanded to 0.2 mm², the system

achieved a power density of 4.32 W/m² (Supplementary Fig. 29 and Table 4). This value not only closely approaches the commercialization benchmark of approximately 5 W/m² but also remains competitive when compared to the reported power density achieved with membranes of the same concentration difference and the same area.”

Supplementary Figure 29. Output power and power density with different test area. Detailed membrane size parameters are shown in Supplementary Table 4. Error bars indicated the standard deviations from three different samples.

Supplementary Table 4. Thickness and width of membrane for different test areas.

Testing area (10 ⁻² mm ²)	Thickness (μm)	Width (cm)
1	2	0.5
2	4	0.5
4	8	0.5
10	20	0.5
20	20	1.0

Reference:

- (44) Pan S, et al. Toward Scalable Nanofluidic Osmotic Power Generation from Hypersaline Water Sources with a Metal–Organic Framework Membrane. *Angew. Chem. Int. Ed.* 62, e202218129 (2023).
- (45) Ding L, Xiao D, Zhao Z, Wei Y, Xue J, Wang H. Ultrathin and Ultrastrong Kevlar Aramid Nanofiber Membranes for Highly Stable Osmotic Energy Conversion. *Adv. Sci.* 9, 2202869 (2022).
- (46) Gao J, Liu X, Jiang Y, Ding L, Jiang L, Guo W. Understanding the Giant Gap between Single-Pore- and Membrane-Based Nanofluidic Osmotic Power Generators. *Small* 15, 1804279 (2019).
- (47) Wang L, Wang Z, Patel SK, Lin S, Elimelech M. Nanopore-Based Power Generation from Salinity Gradient: Why It Is Not Viable. *ACS Nano* 15, 4093-4107 (2021).

Comment 7

About the structural stability of the as-formed membranes, some simple dipping and sonication experiments were not so convincing. Even if the authors stated much more

advantages and superiorities of this work, the practical significance of the research work and the huge economic/energy effects it may bring in the future should be assessed and evaluated.

Response to Comment 7

We thank the reviewer for the kind comment. Point-by-point responses to the comment are enclosed as follows:

1. For the structural stability of Cu-TCPP membrane

As suggested by the reviewer, inductively coupled plasma (ICP) spectrometry was employed to monitor the Cu^{2+} content in the solution during the immersion process of Cu-TCPP membranes. Remarkably, even after prolonged immersion for up to 15 days, the Cu^{2+} content in the solution remained below the detection limit (0.5 mg/L). This indicates that the dissolution of the Cu-TCPP membrane can be essentially disregarded during the immersion process.

2. For the practical significance of our work

To date, the importance of osmotic energy in revolutionizing the global energy landscape has been widely acknowledged. Although researchers have theoretically estimated a substantial 1.0 TW of osmotic energy released globally from the mixing of river water and seawater (*Joule*, 2020, 4, 2244-2248), the practical harnessing of this osmotic energy requires the creation of high and low-concentration solution systems. This necessity poses a significant challenge as it places substantial demands on freshwater or low-concentration solutions, hampering the large-scale practical application of osmotic energy harvesting. In our work, leveraging the exceptional photothermal conversion performance of Cu-TCPP, we have validated that, even in the absence of an ion concentration gradient, selective ion transport behavior can still be achieved. This implies the potential to capture electrical energy from common saline solutions, greatly extending the concept of “osmotic energy” to “ionic energy” and breaking the dependence of traditional osmotic energy harvesting systems on freshwater.

In addition, in the design of our experiment, we also meticulously considered the practical significance of the research from various angles, including scalable

production capabilities and real-world application potential:

1. We opted for the 2D lamellar type of nanofluidic system, known for its tunable nanochannels and the feasibility of large-scale membrane preparation. By enlarging the size of the filtration substrate membrane, we successfully produced a Cu-TCPP membrane with an area of 78.5 cm² (Supplementary Figure 33). Moreover, given the low production efficiency and time-consuming of the vacuum filtration method, in the revised manuscript, we also selected the efficient spraying coating strategy to fabricate a Cu-TCPP membrane with a larger area of 600 cm² (Detailed conditions described in the method and Supplementary Fig. 36), further highlighting the potential for large-scale production and industrial applications of 2D Cu-TCPP membrane.

2. We also explored the feasibility of connecting Cu-TCPP membrane devices in series to ascertain whether the developed nanofluidic system could power practical electronic devices like calculators and LED light bulbs. Inspired by the insightful **Comment 5** as described above, we also assessed the feasibility of osmotic energy harvesting using actual seawater, which furnishes more data for the practical applications of Cu-TCPP membranes.

3. In our investigation of light-responsive ion transport performance, we conducted indoor experiments using a laboratory light source and extended our research to outdoor experiments, demonstrating that light-driven ion transport can be achieved under natural sunlight irradiation. A detailed description has been added in the revised manuscript as follows:

~~“Additionally, large-scale uniform Cu-TCPP membranes could be prepared by changing the filtration area (Supplementary Fig. 19). The membrane structure remained intact even after being immersed in electrolyte solutions with different pH values for 15 days (Supplementary Fig. 20). Furthermore, no disintegration or re-dispersion was observed even under 30 minutes of ultrasonic treatment (Supplementary Fig. 21), indicating that the Cu-TCPP membrane could withstand harsh working conditions.—~~

Large-scale uniform Cu-TCPP membranes with an area of 78.5 cm² were successfully prepared by enlarging the filtration area (Supplementary Fig. 33), and the membrane

structure remained intact even after being immersed in electrolyte solutions with different pH values for 15 days (Supplementary Fig. 34). Inductively coupled plasma (ICP) spectrometry was employed to monitor the Cu^{2+} content in the solution during the immersion process of Cu-TCPP membranes. Remarkably, even after prolonged immersion for up to 15 days, the Cu^{2+} content in the solution remained below the detection limit 0.5 mg/L. Furthermore, no disintegration or re-dispersion was observed even under 30 minutes of ultrasonic treatment (Supplementary Fig. 35), indicating that the Cu-TCPP membrane could withstand harsh working conditions. It is important to note that the production efficiency of the vacuum filtration method is quite low and time-consuming. To overcome this limitation, we employed the spraying coating strategy to fabricate a Cu-TCPP membrane with a larger area of 600 cm^2 at high speed (Detailed conditions described in the method and Supplementary Fig. 36), further highlighting the potential for large-scale production and industrial applications of the 2D Cu-TCPP membrane.

~~The solutions with salinity gradients are a prerequisite for traditional ionic energy recovery systems, as the salinity gradient is the only driving force for directional ion transport.~~ Although researchers have estimated a substantial 1.0 TW of osmotic energy released globally from the mixing of river water and seawater through theoretical calculations⁵², the practical utilization of osmotic energy presents significant challenges due to the requirement for freshwater or low-concentration solutions, because both of the two prevailing technologies for osmotic energy harvesting, RED technology and Pressure Retarded Osmosis (PRO) technology, require solutions of different concentrations on either side of the membrane..

Supplementary Figure 36. Large-scale Cu-TCPP membrane (top: 300 cm², bottom: 600 cm²) fabricated by spraying coating strategy.

Reference:

(52) Logan BE, Elimelech M. Membrane-based processes for sustainable power generation using water. *Nature* 488, 313-319 (2012).

REVIEWER COMMENTS

Reviewer #1 (Remarks to the Author):

My comments have been well addressed.

Reviewer #2 (Remarks to the Author):

I appreciate the author's further explanations in many aspects, such as the comparison with the performance of previous work, the reasons for choosing in-plane channels for performance testing, energy conversion in light-enhanced processes and its differences from previous methods, and issues regarding interface resistance. Firstly, the fundamental energy conversion processes in this work, whether it's salinity gradient conversion or light-enhanced salinity gradient conversion, lack conceptual innovation. Therefore, the most crucial support for this work is the device's performance. The authors also emphasize that the lower device performance compared to previous reports is due to the larger testing area in this work. The author consistently highlights that the synthesis method in this work can rapidly produce larger membrane areas. However, the actual device area used in this work corresponds to the cross-sectional area of the synthesized membrane. When the membrane thickness reaches 20 micrometers (corresponding to a cross-sectional area of 10^{-2} mm^2), the output power of the device has already declined to 4W. This area difference, compared to practical device applications and existing works, is not significant and falls far away from macroscopic device. The key issue is that this device configuration (using the cross-sectional area of the membrane) does not have an advantage in amplifying the device compared to traditional configurations. Therefore, in terms of device performance, I still do not believe that this work significantly improves upon existing efforts. Consequently, I still do not think this work suitable for publication in Nature Communications.

Reviewer #3 (Remarks to the Author):

This manuscript has been greatly improved after the revision and could be accepted for publication.

Review #1

General comment

My comments have been well addressed.

Response to the General Comment

We greatly thank the reviewer for the recommendation to accept the paper for publication.

Review #2

General comment

I appreciate the author's further explanations in many aspects, such as the comparison with the performance of previous work, the reasons for choosing in-plane channels for performance testing, energy conversion in light-enhanced processes and its differences from previous methods, and issues regarding interface resistance. Firstly, the fundamental energy conversion processes in this work, whether it's salinity gradient conversion or light-enhanced salinity gradient conversion, lack conceptual innovation. Therefore, the most crucial support for this work is the device's performance. The authors also emphasize that the lower device performance compared to previous reports is due to the larger testing area in this work. The author consistently highlights that the synthesis method in this work can rapidly produce larger membrane areas. However, the actual device area used in this work corresponds to the cross-sectional area of the synthesized membrane. When the membrane thickness reaches 20 micrometers (corresponding to a cross-sectional area of 10^{-2} mm^2), the output power of the device has already declined to 4W. This area difference, compared to practical device applications and existing works, is not significant and falls far away from macroscopic device. The key issue is that this device configuration (using the cross-sectional area of the membrane) does not have an advantage in amplifying the device compared to traditional configurations. Therefore, in terms of device performance, I still do not believe that this work significantly improves upon existing efforts. Consequently, I still do not think this work suitable for publication in Nature Communications.

Response to the General Comment

We express our gratitude for the meticulous review process undertaken by the esteemed reviewer, whose insightful comments have significantly contributed to the refinement of our work. Inspired by the kind suggestions in the initial revision, the potential of the membrane in large-scale preparation and working conditions was further developed. Upon careful consideration of the above questions raised by the reviewers, we fully acknowledge the

importance of discussing these issues, and the point-to-point response to the comments is as follows:

1. The author consistently highlights that the synthesis method in this work can rapidly produce larger membrane areas. However, the actual device area used in this work corresponds to the cross-sectional area of the synthesized membrane.

As described in the first round of revision, one of the advantages of two-dimensional nanochannel membranes is their potential for large-scale production. In the chosen lateral diffusion model, membrane thickness is a crucial parameter as kindly suggested by the reviewer. While it also should be noted that the other geometric parameters of the membrane, including length and width, also need to be carefully design. For example, to maximize the impact of light illumination, establishing a temperature gradient within the membrane confirms the importance of membrane length as shown in Fig. 4g. Therefore, the rapid preparation of large-area membranes is crucial for the scale-up of 2D MOF membrane-based nanofluidic devices.

2. When the membrane thickness reaches 20 micrometers (corresponding to a cross-sectional area of 10^{-2} mm^2), the output power of the device has already declined to 4W. This area difference, compared to practical device applications and existing works, is not significant and falls far away from macroscopic device.

Inspired by the comments suggested in the first-round revision, the output power density of membranes with increasing testing area was explored. First, we respectfully bring to the reviewer's attention an oversight in the assessment. According to our experimental results, when maintaining the output power density at 4.3 W/m^2 , the membrane testing area increased to 0.2 mm^2 (Supplementary Figure 29), not the 0.01 mm^2 as mentioned in the comment. As we know, in the current research landscape of osmotic energy recovery, maintaining a high level of power density as the testing area increases is a significant challenge. Based on our literature investigation as shown in Table R1, under the constant testing area of 0.2 mm^2 and the same salinity gradient, only very few studies reported superior power densities.

In the revised manuscript, the power density of membrane with larger testing area was also further investigated. We confirmed that the power density of the system showed little decrease and remained relatively stable around 4 W/m^2 and the power density would reach a plateau when the membrane area was enlarged from 0.2 mm^2 to 0.6 mm^2 . Such a changing tendency could be observed in many previous reports (*J. Am. Chem. Soc.*, 2022, 144, 12400-12409; *Adv. Funct. Mater.*, 2023, 2302427; *Angew. Chem. Int. Ed.*, 2023, 62, e202218129). The previous reports proposed that the relationship between the power density and the testing area could be demonstrated by the following equation (*Angew. Chem. Int. Ed.*, 2021, 133, 20456-20462; *Nano Energy*, 2022, 104, 107981):

$$\rho_m = \frac{P_m}{S} = \frac{1}{4} \times \frac{(1 - \alpha)^2 U_0^2}{(R_I + R_{II})S + K_m}$$

In the equation, ρ_m is the maximum power density, S represents the test area, U_0 represents the open-circuit voltage, α is the loss coefficient due to the polarization potential, R_I is the internal resistance of the test device, and K_m represents the ion transport resistance factors across the gradient membrane. Based on these studies, under a fixed concentration gradient, R_I and R_{II} are kept constant for a given testing device. As a result, except for the U_0 , and K_m determined by ion-selective membranes, the output power of the RED device inversely correlates with the testing area, ultimately stabilizing at a low value (*Nano Energy*, 2022, 104, 107981). Furthermore, as discussed in the first-round revision, when developing from single-pore to membrane-scale application, it is crucial not to simply ignore the reservoir/membrane interfacial resistance due to the concentration polarization phenomenon, as it also contributes to a decrease in power density with an increasing test area (*Small*, 2019, 15, 1804279). As the pore density continuously increased, the concentration polarization zone of every channel will collapse into one joint zone, and the interfacial resistance kept constant. Additionally, Elimelech et al. proposed another important perspective, stating that unavoidable concentration polarization would result in a decreased membrane potential and ionic conductance due to the lower transmembrane concentration gradient and decreasing driving force for ion diffusion (*ACS Nano*, 2021, 15, 4093-4107). While the mechanism governing the relationship between energy performance and membrane area remains complex, it should be noted that the power density of the Cu-TCPP membrane stabilized at 3.9 W/m² with a testing area of 0.6 mm², surpassing the reported power density achieved with most membranes of the same concentration difference and testing area. As of now, to the best of our knowledge, only one reported COF membrane showed a higher power density (*J. Am. Chem. Soc.*, 2022, 144, 12400-12409). However, the scalable production of this COF membrane is difficult, as we described in the first-round revision

Furthermore, we also attempted to introduce light illumination into the energy conversion system with an enlarged testing area. The enhanced ion transport driving force, attributed to the temperature difference in the membrane, led to the membrane's power density reaching 8.27 W/m² with a testing area of 0.6 mm², surpassing the commercialization benchmark at such a testing area and salinity gradient. This achievement is rarely reported, underscoring the importance of our work in osmotic energy conversion studies.

3. The key issue is that this device configuration (using the cross-sectional area of the membrane) does not have an advantage in amplifying the device compared to traditional configurations.

As described in the initial revision, in studies on two-dimensional nanochannel membranes, the lateral diffusion mode has recently garnered increasing attention because it could unleash the full potential of interlayer channels in ion sieving (*Nat. Nanotech.*, 2017, 12, 546-550; *Angew. Chem. Int. Ed.*, 2020, 132, 8798-8804). Alongside this research enthusiasm, various device preparation methods have also been reported. For instance, Cao et al. reported the use of UV gel for MXene membrane embedding, which could significantly shorten the drying time and enhance device production efficiency (*ACS Nano*, 2020, 14, 16654-16662). We believe that, with more research efforts being invested, more efficient scaling-up strategies will be developed in future studies on lateral 2D membrane.

Furthermore, to address the reviewer's concerns about device scaling, additional experiments were also conducted. Multiple membranes were facilely assembled laterally to produce a larger osmotic energy recovery device. The results indicate that, with an increase in the number of membranes, the stable ionic current exhibited a nearly linear increase, while the power density remained relatively constant. This result primarily confirms the feasibility of scaling up the energy conversion device based on the Cu-TCPP membrane. We apologize to the reviewers if the experiments so far were inadequate. In any case, the valuable comment is a great inspire for us to conduct more in-depth studies on the practical osmotic energy conversion system construction in future works.

The detailed experimental data have been added to the revised manuscript and Supplementary Materials as follows:

~~Furthermore, as the membrane thickness and corresponding testing area increased, we observed a rise in output power attributed to the heightened ion flux. However, due to the substantial accumulation of K^+ ions at the channel exit and the onset of concentration polarization in thicker membranes, a decrease in power density was also noted, consistent with findings from prior studies⁴⁴⁻⁴⁶. Recent reports have indicated this concentration polarization has even been identified as a significant obstacle to the practical implementation of RED technology⁴⁷. It should be noted that when the membrane thickness reached 20 μm and the test area expanded to 0.2 mm^2 , the system achieved a power density of 4.32 W/m^2 (Supplementary Fig. 29 and Table 4). This value not only closely approaches the commercialization benchmark of approximately 5 W/m^2 but also remains competitive when compared to the reported power density achieved with membranes of the same concentration difference and the same area.~~

Currently, in the field of osmotic energy harvesting, the inverse relationship between power density and increasing testing area is a critical challenge⁴⁴. For the Cu-TCPP membrane, as the testing area increased from 0.01 to 0.2 mm^2 , there was a sharp decrease in output density (Supplementary Fig. 29 and Table 4). As the testing area further increased, similar to the

previous reports, the trend of the power density becomes more gradual. Previous reports^{45,46} proposed that the relationship between the power density and the testing area could be demonstrated by the following equation:

$$\rho_m = \frac{P_m}{S} = \frac{1}{4} \times \frac{(1 - \alpha)^2 U_0^2}{(R_I + R_{II})S + K_m}$$

In the equation, ρ_m is the maximum power density, S represents the test area, U_0 represents the open-circuit voltage, α is the loss coefficient due to the polarization potential, R_T is the internal resistance of the test device, and K_m represents the ion transport resistance factors across the gradient membrane. Based on these studies, under a fixed concentration gradient, R_I and R_{II} are kept constant for a given testing device. As a result, except for the U_0 , and K_m determined by ion-selective membranes, the output power of the RED device inversely correlates with the testing area, ultimately stabilizing at a low value. Moreover, Gao et al. have indicated that the concentration polarization phenomenon, caused by the substantial accumulation of K^+ ions at the channel exit as the testing area expands, will occur during the energy conversion process. Consequently, the reservoir/membrane interfacial resistance should not be ignored, leading to a decrease in power density with an increasing test area⁴⁷. Elimelech et al. suggested that the unavoidable concentration polarization would also result in a decreased membrane potential and ionic conductance due to the lower transmembrane concentration gradient and decreasing driving force for ion diffusion⁴⁸. It should be noted that the power density of Cu-TCPP membrane stabilized at 3.9 W/m² with the testing area of 0.6 mm², which is superior to most the reported power density achieved with membranes of the same concentration difference and testing area. Furthermore, with the assistance of light irradiation, the ion diffusion driven by the synergistic effect of photo-thermal/concentration gradient led to a further increase in the power density of the Cu-TCPP membrane. When the test area reached 0.6 mm², the power density of the system achieved 8.27 W/m², surpassing the commercialization benchmark of 5 W/m².

To assess the scalability of our Cu-TCPP membrane-based device and extend the osmotic energy recovery system, various numbers of Cu-TCPP membranes were assembled into an enlarged experimental setup. As depicted in Supplementary Fig. 58, a single Cu-TCPP-RED demonstrated a short-circuit current of 17.2 μ A in an artificial seawater/river setup. For the Cu-TCPP-RED devices comprising 1 to 8 membrane units, the output current exhibited nearly linear growth in correlation with the number of membrane units, suggesting an increase in total ion flux with the expanded ion transport area. The output current also demonstrated excellent stability with negligible fluctuations over a duration of 7200 seconds (Supplementary Fig. 59).

Furthermore, the power density remained relatively stable despite the increase in the number of membrane units in the system, affirming the practical application potential of 2D Cu-TCPP membranes in osmotic energy harvesting.

Supplementary Figure 29. Power density with different test area. Detailed membrane size parameters were shown in Supplementary Table 4. Error bars indicated the standard deviations from three different samples.

Supplementary Figure 30. Output power of different test areas under illuminated and non-illuminated conditions.

Supplementary Figure 58. Relationship between current, power density and number of Cu-TCPP-RED membrane units. Error bars indicated the standard deviations from three different samples.

Supplementary Figure 59. Long-term open-circuit ionic currents in different numbers of Cu-TCPP membranes.

Table 4. Thickness and width of membrane for different test areas.

Testing area (10^{-2} mm ²)	Thickness (μ m)	Width (cm)
1	2	0.5
2	4	0.5
4	8	0.5
10	20	0.5
20	20	1.0
40	20	2.0
60	20	3.0

Table R1. Comparison of osmotic energy conversion performance with partially reported nanofluidic membranes.

Membrane	Test area (mm ²)	Salinity gradient	Power density (W/m ²)	Ref.
All-polysaccharide poly electrolyte hydrogel membranes	0.2	50	about 1.5	1
Ti ₃ C ₂ T _x Mxene membranes	0.2	50	4.6	2
Graphene oxide membrane	0.8	50	0.77	3
Bacterial cellulose membranes	0.18	50	0.23	4
Polymeric-C ₃ N ₄ membrane	0.15	1000	0.21	5
BN nanopore membrane	0.1	1000	0.17	6
Ultrathin silica isoporous membrane	0.25	50	0.024	7
Nanokaolinite membranes	0.2	100	0.18	8
Mxene/GO membrane	0.2	50	1.5	9
p-MOF-AAO membrane	0.1256	50	0.97	10
PSS/HKUST-1 membrane	0.2	50	1.8	11
VNMs/CoAl-LDH membrane	1.6	50	0.6	11
VNMs/CoAl-LDH membrane	0.2	50	0.4	12
NcCNC/GO pair membrane	0.2	50	3.0	13

GO membrane	0.2	50	18	
ChCNC/GO membrane	0.2	50	4.73	
GO-CA membrane	0.2	50	0.13	14
PI membrane	0.2	50	0.39	15
MMT membrane	0.2	50	0.7	16
COF/AAO	0.28	50	0.53	17
BC-CMC	0.25	50	2.25	18
COF-SO ₃ H-24 and COF-QA-24 membranes	0.8	50	19	19
	0.2	50	4.32	
Cu-TCPP membrane (Without light irradiation)	0.4	50	4.09	
	0.6	50	3.90	
	0.2	50	9.42	This work
Cu-TCPP membrane (With light irradiation)	0.4	50	8.64	
	0.6	50	8.27	

Reference:

(1) Bian G, *et al. Angew. Chem. Int. Ed.* **60**, 20294-20300 (2021); (2) Ding L, *et al. Angew. Chem. Int. Ed.* **59**, 8720-8726 (2020); (3) Ji J, *et al. Adv. Funct. Mater.* **27**, 1603623 (2017); (4) Sheng N, *et al. ACS Appl. Mater. Inter.* **13**, 22416-22425 (2021); (5) Xiao K, *et al. Angew. Chem. Int. Ed.* **57**, 10123-10126 (2018); (6) Pendse A, *et al. Adv. Funct. Mater.* **31**, 2009586 (2021); (7) Yan F, *et al. J. Mater. Chem. A* **7**, 2385-2391 (2019); (8) Cheng H, *et al. Adv. Mater.* **29**, 1700177 (2017); (9) Wang F, *et al. J. Membrane Sci.* **647**, 120280 (2022); (10) Li ZQ, *et al. Angew. Chem. Int. Ed.* **61**, e202202698 (2022); (11) Pan S, *et al. Angew. Chem. Int. Ed.* **62**, e202218129 (2023); (12) Yu X, *et al. Small*, 2300338 (2023); (13) Zhao W, *et al. Nano Energy* **98**, 107291 (2022); (14) Jia P, *et al. Molecules* **26**, 5343 (2021); (15) Sun Y, *et al. Nano Energy* **105**, 108011 (2023); (16) Wei C, *et al. J. Membrane Sci.* **680**, 121744 (2023); (17) Chen M, *et al. Adv. Funct. Mater.* 2302427 (2023); (18) Wu Z, *et al.* **88**, 106275 (2021); (19) Cao L, *et al. J. Am. Chem. Soc.* **144**, 12400-12409 (2022).

Review #3

General comment

This manuscript has been greatly improved after the revision and could be accepted for publication.

Response to the General Comment

We greatly thank the reviewer for the positive comment.

REVIEWERS' COMMENTS

Reviewer #2 (Remarks to the Author):

My concerns have been addressed.

Review #2

General comment

My concerns have been addressed.

Response to the General Comment

Thank you for your valuable suggestions. We are delighted to be able to address your concerns as well. Thank you again for your kind reminders during the review period.